# ATP synthase evolution on a cross-braced dated tree of life

Tara A. Mahendrarajah[1], Edmund R. R. Moody[2,3,10], Dominik Schrempf[4,5,10], Lénárd L. Szánthó[4,5,6], Nina Dombrowski[1], Adrián A. Davín[7], Davide Pisani[2,3], Philip C. J. Donoghue[3], Gergely J. Szöllősi[4,5,8], Tom A. Williams[2] ✉ & Anja Spang[1,9] ✉

The timing of early cellular evolution, from the divergence of Archaea and Bacteria to the origin of eukaryotes, is poorly constrained. The ATP synthase complex is thought to have originated prior to the Last Universal Common Ancestor (LUCA) and analyses of ATP synthase genes, together with ribosomes, have played a key role in inferring and rooting the tree of life. We reconstruct the evolutionary history of ATP synthases using an expanded taxon sampling set and develop a phylogenetic cross-bracing approach, constraining equivalent speciation nodes to be contemporaneous, based on the phylogenetic imprint of endosymbioses and ancient gene duplications. This approach results in a highly resolved, dated species tree and establishes an absolute timeline for ATP synthase evolution. Our analyses show that the divergence of ATP synthase into F- and A/V-type lineages was a very early event in cellular evolution dating back to more than 4 Ga, potentially predating the diversification of Archaea and Bacteria. Our cross-braced, dated tree of life also provides insight into more recent evolutionary transitions including eukaryogenesis, showing that the eukaryotic nuclear and mitochondrial lineages diverged from their closest archaeal (2.67-2.19 Ga) and bacterial (2.58-2.12 Ga) relatives at approximately the same time, with a slightly longer nuclear stem-lineage.

The phylogeny and timeline of early cellular evolution, including the age of the last universal common ancestor (LUCA), the radiations of the archaeal and bacterial domains, and the origin of eukaryotes and their progenitor prokaryote lineages, is poorly constrained[1]. Recent genomics approaches have greatly improved our sampling of natural diversity and uncovered previously unknown microbial lineages that are key to understanding early cellular evolution[2]. For instance, the Asgard archaea[3–5] (also referred to as Asgardarchaeota[6], Supplementary Data 1), include the closest known sister lineage of the Eukaryota (i.e. eukaryotes)[3,4,7,8] and have provided support for the evolution of the eukaryotic cell through a symbiosis between at least one asgardarchaeal and one alphaproteobacterial partner[9–15]. The

[1]Department of Marine Microbiology and Biogeochemistry, NIOZ, Royal Netherlands Institute for Sea Research, AB Den Burg, The Netherlands. [2]Bristol Palaeobiology Group, School of Biological Sciences, University of Bristol, BS8 1TQ Bristol, UK. [3]Bristol Palaeobiology Group, School of Earth Sciences, University of Bristol, BS8 1TQ Bristol, UK. [4]Department Biological Physics, Eötvös University, Pázmány P. stny. 1A., H-1117 Budapest, Hungary. [5]MTA-ELTE "Lendulet" Evolutionary Genomics Research Group, Pázmány P. stny. 1A., H-1117 Budapest, Hungary. [6]Institute of Evolution, Centre for Ecological Research, Karolina ut 29, H-1113 Budapest, Hungary. [7]Department of Biological Sciences, Graduate School of Science, The University of Tokyo, Tokyo, Japan. [8]Model-Based Evolutionary Genomics Unit, Okinawa Institute of Science and Technology Graduate University, Okinawa, Japan. [9]Department of Evolutionary & Population Biology, Institute for Biodiversity and Ecosystem Dynamics (IBED), University of Amsterdam, Amsterdam, The Netherlands. [10]These authors contributed equally: Edmund R. R. Moody, Dominik Schrempf. ✉e-mail: tom.a.williams@bristol.ac.uk; anja.spang@nioz.nl

discovery of the symbiotic and genome-reduced members of the bacterial Candidate Phyla Radiation (CPR) and the DPANN archaea (named after the first member lineages of this group, the Diapherotrites, Parvarchaeota, Aenigmarchaeota, Nanoarchaeaota, and Nanohaloarchaeota)[16–18], that were originally interpreted as early diverging branches on each side of the root of the tree of life[2], might be important for our understanding of the deep split separating Archaea and Bacteria[19]. However, more recent phylogenomic analyses suggest that CPR are instead sister to Chloroflexota within Terrabacteria[19–23]. Time scaling molecular evolution is challenging because the rate of molecular evolution has varied substantially through time[19,24–27] and, with few fossil calibrations (e.g. maximum age constraints and lack of Precambrian maximum age calibrations), clock models struggle to capture this rate variation. This has led to estimates of divergence time that in some cases are uncertain (as in the case of the age of LUCA – 4.52–4.48 Ga[19,27,28]). Additional sources of temporal information beyond fossil and geochemical calibrations are crucial to improve these estimates of divergence time.

The ATP synthase is a protein complex central to energy conservation through the synthesis and hydrolysis of ATP[29,30]. It is a useful marker to address key evolutionary transitions due to the presence of this enzyme across all domains of life[24,29,31–37]. The ATP synthase family is classified into the F-, A-, and V-type ATP synthases[30,33,34] based on taxonomic affiliation, function, and cellular localization[30,34,38]. F-type ATP synthases are ubiquitous across bacteria and eukaryotes and localize to cellular, mitochondrial, and plastid membranes[39]. In line with this, eukaryotic F-type ATP synthases are hypothesized to be derived from the bacterial ancestors of these organelles[33,35,40]. The A-type ATP synthase[34,38,41], found primarily in Archaea, belongs to a larger family of A/V-type ATP synthases[38] that also include eukaryotic complexes found in vacuoles[32–34,38,40–42]. The F- and A/V-type ATP synthases share a common foundational architecture consisting of a soluble cytoplasmic component (R1) connected to an insoluble membrane component (R0) (Supplementary Fig. 1). The hexameric headpiece of the R1 complex contains three copies each of a catalytic (c) and non-catalytic (nc) subunit and is the site of ATP synthesis and hydrolysis. The catalytic and non-catalytic subunits comprising the soluble hetero-hexameric R1 component, are paralogs to each other. They arose prior to LUCA through an ancient duplication of a RecA family protein (P-loop NTPase) followed by the loss of the catalytic function in one subunit[29,32–35,38,40,43]. Due to this ancient gene duplication, each paralog can act as an outgroup to the other, providing a way to root the tree of life[44]. Since the duplication occurred before the divergence between Archaea and Bacteria, speciation events during the subsequent history of life appear at least twice in the ATP synthase gene tree. This circumstance has been used to improve date estimates for eukaryotic evolution by "cross-bracing" (constraining to the same unknown age) equivalent speciation nodes in the gene tree[24], which propagates limited fossil evidence across the tree. Cross-bracing improves clock estimates in two ways. First, the information that two distant nodes in the tree must have the same age provides a useful constraint on the rates of evolution and ages of the intervening branches. Second, it enables the fossil record of eukaryotes to inform divergence times within both the archaeal and bacterial domains, because eukaryotes obtained ATP synthase paralogs from both sources. In principle, this approach might be expanded beyond ATP synthase to the core set of ribosomal proteins that are conserved between the nucleus, mitochondrion, and chloroplast of eukaryotes as a result of the mitochondrial and plastid endosymbioses. Cross-bracing a ribosomal species tree could help to avoid difficulties arising from HGT events during ATP synthase evolution[32,33,40,45], and the limited resolving power of single gene trees.

To improve our understanding of the evolutionary history of ATP synthases and cellular evolution, we perform phylogenetic analyses using an updated taxon sampling set, ancestral sequence reconstruction[46,47], and novel molecular dating approaches including cross-bracing[24,48,49]. We also use probabilistic gene- and species-tree reconciliation methods (implemented in Amalgamated Likelihood Estimation (ALE))[50–52] to determine the origin and evolution of the ATP synthase and each of its subfamilies. ALE allows us to compare the ATP synthase gene family tree to the tree of life and to infer the history of gene duplication, transfer, and loss during ATP synthase evolution. We assemble a set of ribosomal marker proteins that includes three distinct clades of eukaryotic homologs derived from archaeal, alphaproteobacterial, and cyanobacterial ancestors. Due to gene duplications (ATP synthase) and endosymbiosis events (ribosomal marker genes and ATP synthase), cross-bracing can be applied to both datasets. Our analyses confirm the split of the catalytic and non-catalytic ATP synthase subunits prior to LUCA and reveal the prevalence and early evolution of A/V-type ATP synthases in Bacteria. Our dating analyses establish absolute age estimates for LUCA, the Last Bacterial Common Ancestor (LBCA), and the Last Archaeal Common Ancestor (LACA). We link early cellular evolution with the origin of the head component of ATP synthases, which has diversified earlier than previously assumed. Finally, our analyses improve time estimates for the origin of eukaryotes from its prokaryotic ancestors and thereby inform on eukaryogenesis.

## Results

### Distribution of ATP synthases across Bacteria, Archaea and Eukaryotes

We analysed the distribution of ATP synthase genes across our reference genome dataset of 800 Archaea, Bacteria, and eukaryotes (Figs. 1–2, Supplementary Fig. 2, Supplementary Data 1–4). In agreement with previous work[29,32–36,38,40], our results indicate a partitioning of the F- and A/V-type ATP synthases by domain. Archaea and Bacteria contain primarily A/V-type and F-type subunits, respectively, and eukaryotes harbor complexes of both types (Fig. 1, Supplementary Fig. 2). However, in Bacteria the pattern is more complex than generally assumed[33,35,53]. Consistent with emerging evidence that several bacteria contain A/V-type ATP synthases[35], we found that 46% (23/50) of bacterial phylum-level lineages encode genes for A/V-type ATP synthases in conjunction with (n = 19), or to the exclusion of (n = 4), a bacterial F-type ATP synthase (Fig. 1, Supplementary Fig. 2, Supplementary Data 4). Conversely, only three members of a single archaeal lineage, Methanosarcinales, contain F-type ATP synthases in addition to their A/V-type complex (Fig. 1, Supplementary Fig. 2, Supplementary Data 4), as observed previously[54–56].

Despite the core role of ATP synthases in energy conservation, some prokaryotes (including members of the DPANN archaea[57–59]) lack functional homologs (Fig. 1, Supplementary Fig. 2, Supplementary Data 4). These absences are present across related lineages, suggesting a genuine loss, rather than metagenome-assembled genome incompleteness. Other DPANN lineages, such as Nanohaloarchaeota, may have inherited their ATP synthase from a DPANN ancestor (Supplementary Fig. 6) or acquired ATP synthase genes from symbiotic partners (Supplementary Fig. 7–10, Discussion)[45,60]. Several Bathyarchaeota lack ATP synthase complexes (Fig. 1, Supplementary Fig. 2, Supplementary Data 4), consistent with the previously noted absence of ATP synthases in the Bathyarchaeote BA1 and BA2[61,62]. Instead, these organisms may produce ATP through substrate-level phosphorylation using a putative ATP-forming acetyl-CoA synthetase[61,62]. We found that 60.6% (20/33) of the analysed CPR encode F-type ATP synthases, with phylogenetic trees suggesting inheritance from a common ancestor with Chloroflexi (Fig. 1, Supplementary Fig. 2, Supplementary Data 4). Conversely, 30.3% (10/33) of the sampled CPR have lost genes that would enable the formation of a canonical ATP synthase complex (Supplementary Fig. 2, Supplementary Data 4). Interestingly, three members of the CPR in our dataset lack an F-type ATP synthase but have a near-complete or complete A/V-type complex

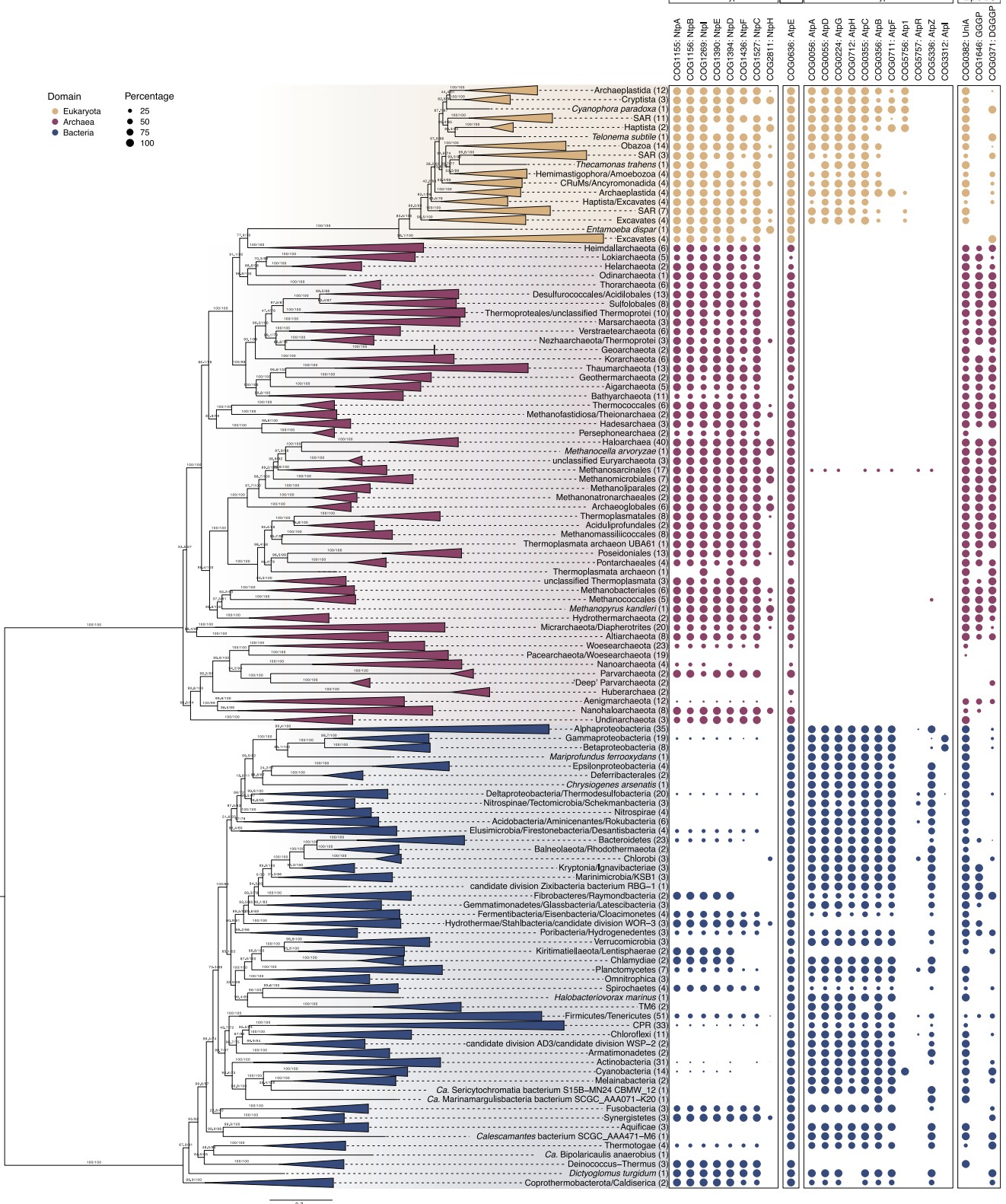

**Fig. 1 | Distribution of COG families representing the F- and A/V-type ATP synthase subunits and select lipid biosynthesis genes across the tree of life.** COG families corresponding to the ATP synthase subunits and lipid biosynthesis genes (see Methods for selection of COG families, Supplementary Data 3) are represented as a percentage presence by phylogenetic cluster, consistent with collapsed taxonomic clades in the maximum-likelihood concatenated species tree. The concatenated alignment contains 780 taxa and was trimmed with BMGE v1.12 (settings: -m BLOSUM30 -h 0.55)[110] to remove poorly-aligning positions (final alignment length = 3367 amino acids). The maximum-likelihood tree was inferred using IQ-TREE2 v2.1.2 with the LG+C20+R+F model with SH-like approximate likelihood (left) and ultrafast bootstrap approximation (right), each with 1000 replicates[111,122,123]. The scale bar corresponds to the expected number of substitutions per site. Color code: archaea = red, bacteria = blue, eukaryotes = yellow.

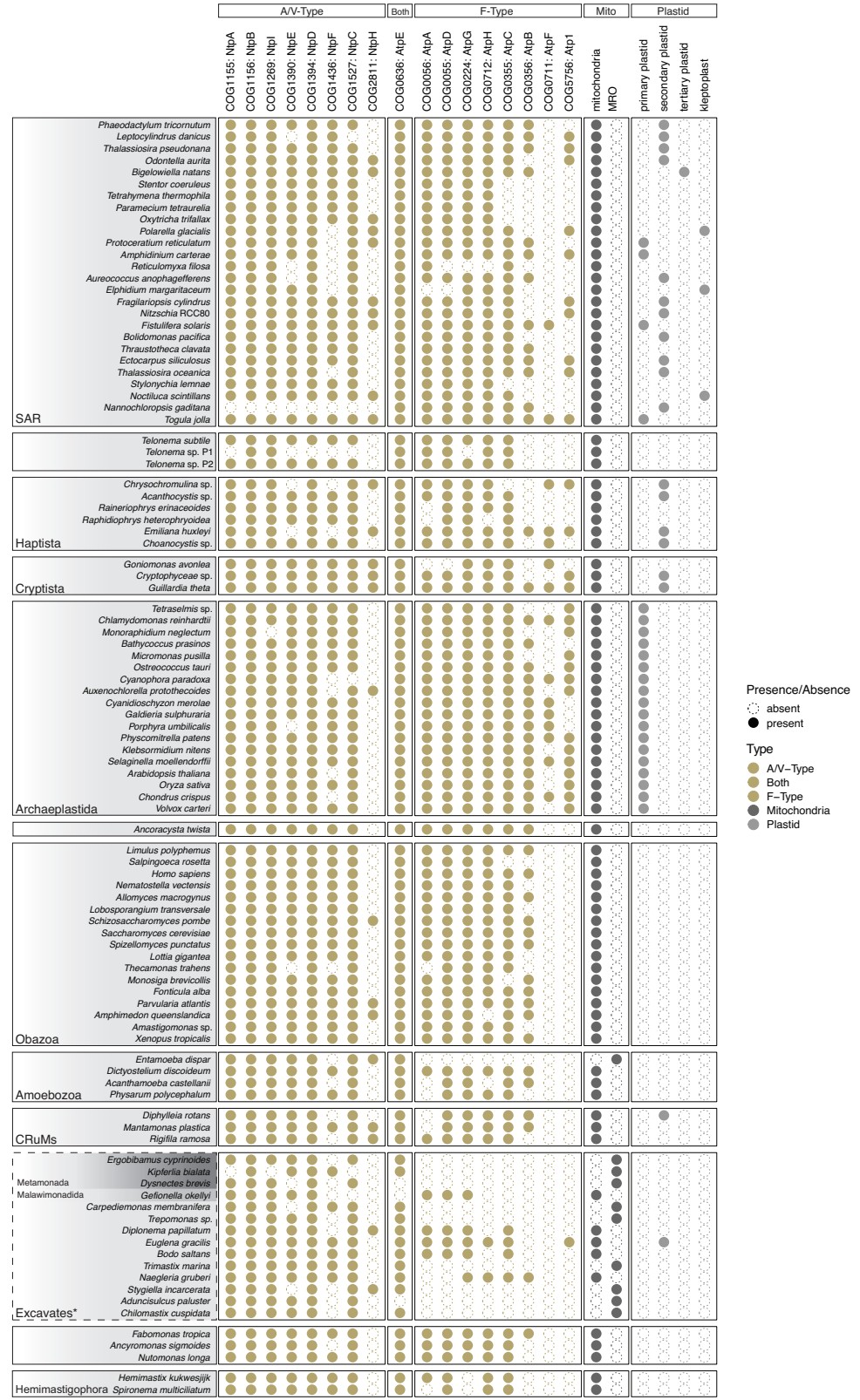

**Fig. 2 | Occurrence of COG families representing the F- and A/V-type ATP synthase subunits and the presence/absence of key metabolic organelles across the 100 sampled eukaryotes.** COG families representing the ATP synthase subunits (see Methods for selection of COG families, Supplementary Data 3) are presented as binary presence-absence counts per taxon. The relationships among eukaryotic supergroups is consistent with Burki, 2020[136]. Dashed lines represent groups with greater uncertainty. Mito = mitochondria and mitochondrion-related organelles (MROs). Plastid = primary-, secondary-, and tertiary-plastid, and kleptoplast. See Supplementary Data 5 for additional information on organelle distribution. The list of eukaryotic ATP synthase sequences flagged as putative bacterial contamination can be found in Supplementary Data 4.

(Fig. 1, Supplementary Fig. 2), indicating a recent acquisition of the A/V-type complex via HGT potentially by members of the Synergistetes (Supplementary Fig. 2, 6–10).

Most eukaryotic lineages contained core functional subunits of both F- and A/V-type ATP synthases, with the exception of 24/100 analyzed representatives, including *Entamoeba dispar* and certain Excavates (Fig. 2, Supplementary Data 5). This is consistent with the energy metabolism of these anaerobes whose mitochondrion-related organelles have lost components of the aerobic electron transport chain[63,64] (Fig. 2). We observed that 78% (14/18) of Archaeplastida encode genes for Atp1, the F-type alpha subunit of cyanobacteria (COG5756, Fig. 2, Supplementary Data 5). Notably, this gene is lacking in species without a plastid (Fig. 2), consistent with the existence of a second F-type ATP synthase of endosymbiotic origin in chloroplasts. However, 36% of plastid-bearing eukaryotes (16/45) lack an Atp1 homolog (Fig. 2), indicating subsequent loss of Atp1 in some photo-synthetic eukaryotes.

### Evolutionary history of soluble ATP synthase subunits

Phylogenetic analyses including all catalytic (*c*) and non-catalytic (*nc*) subunits of the soluble head component of the F- and A/V-type ATP synthase (Supplementary Fig. 1), the F1 beta and A1/V1A and the F1 alpha and A1/V1B, respectively (hereafter referred to as *c*F1, *c*A1V1, *nc*F1, *nc*A1V1), revealed four clades corresponding to each of the four protein families (Fig. 3A, Supplementary Fig. 10). Based on the gene family tree and the observation that all organisms encoding an ATP synthase possess catalytic (*c*F1 and *c*A1V1) (Fig. 3A, Supplementary Figs. 4, 6, 8–10) and non-catalytic (*nc*F1 and *nc*A1V1) (Fig. 3A, Supplementary Figs. 5, 7–10) subunits, our analysis agrees with the consensus view[31,32,35,40], that the deepest split lies between those families (Fig. 3A, Supplementary Fig. 10)[31,32,35,40]. Our results suggest an early divergence of the functional capacities of each subunit followed by subsequent bifurcations into F- and A/V-type complexes (Fig. 3A, Supplementary Fig. 10). The deep splits observed within each of the catalytic and non-catalytic subunits of the F- and A/V-type complexes have been

hypothesized to coincide with the speciation of Archaea and Bacteria[31,32,65] (Fig. 3A).

Determining which of the four head-forming subunits was present in LACA, LBCA, and LUCA based on gene tree inspection is challenging. For instance, the identification of A/V-type ATP synthases in many Bacteria (Fig. 1, Supplementary Fig. 2) and the recent inference of the presence of components of both F- and A/V-Type ATP synthases in the genome of LBCA[21], challenge a late horizontal acquisition of the A/V-type ATP synthase by Bacteria. To evaluate these hypotheses within a statistical framework, we used the ALE probabilistic approach[51] to reconcile gene trees for each of the ATP synthase subunits with the species tree as a whole, using distinct data treatments (Methods, Supplementary Data 6). This approach compares the gene family tree with the species tree to infer gene origination, duplication, transfer, and loss events. It maps branches of the gene tree to the species tree, using conditional clade probabilities[66] to account for uncertainty in the gene family tree analyses[51]. These analyses agreed with our manual inspection of the gene trees, suggesting that the *c*A1V1 and *nc*A1V1 subunits were present in LACA (presence probability, PPs = 0.99–1)[67] and the *c*F1 and *nc*F1 subunits were present in LBCA (PPs = 0.99–1)[21]. We recovered support for the presence of the *c*A1V1 (PPs = 0.64–1) subunit in LBCA, as has been suggested recently[21], while the presence of the *nc*A1V1 subunit in LBCA was supported only in trees inferred using the LG+C20+R+F but not LG+C60+R+F model (C20: PPs = 0.99–1, C60: PPs = 0.21–0.28, Supplementary Data 6).

The *nc*F1 (PPs = 0.79–1), *nc*A1V1 (PPs = 0.99–1), *c*F1 (PPs = 1), and *c*A1V1 (PPs = 0.99–1) gene families were estimated to having been present in LUCA, suggesting a putative pre-LUCA duplication of both the catalytic and non-catalytic subunits into the F- and A/V-type lineages (Supplementary Data 6, Fig. 4). However, deep branches in the gene trees are susceptible to systematic error, and distinguishing ancestral presence from early horizontal acquisition is difficult[21]. Nonetheless, the widespread presence of genes encoding A/V-type subunits in modern Bacteria (Fig. 1, Supplementary Fig. 2, Supplementary Data 4) suggests that these genes were acquired early in bacterial evolution.

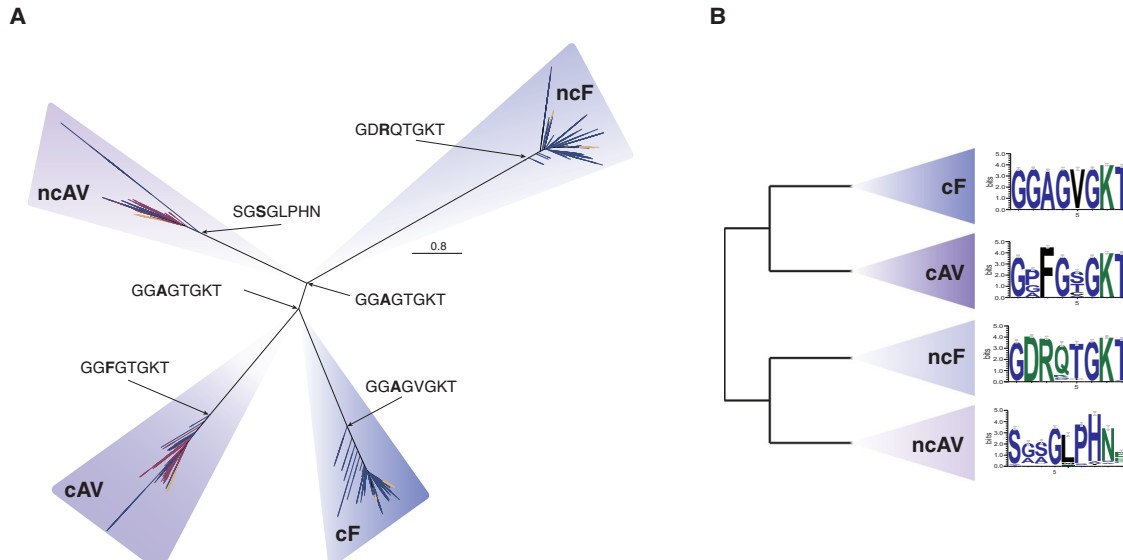

**Fig. 3 | Maximum-likelihood tree of all ATP synthase headpiece subunits identified in sampled Archaea (red), Bacteria (blue), and Eukaryotes (yellow). A** Homologs corresponding to each subunit form monophyletic clusters for each protein family. Catalytic subunits (*c*F1 and *c*A1V1) and non-catalytic subunits (*nc*F1 and *nc*A1V1) cluster together on either side of the root. The alignment contains 1520 sequences and was trimmed with BMGE v1.12 (settings: -m BLOSUM30

-h 0.55)[110] (alignment length = 350 amino acids). The maximum-likelihood tree was inferred using IQ-TREE2 v2.1.2 with the LG+C50+R+F model, selected using the best-fitting model (chosen by BIC)[111,123,132]. The scale bar corresponds to the expected number of substitutions per site. The Walker-A motif from ancestrally reconstructed sequences[123] are shown at their respective nodes. **B** Conserved protein motifs for each subunit derived from the same alignment.

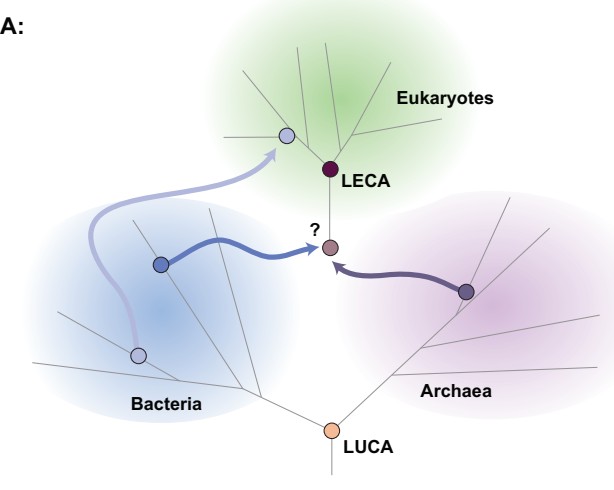

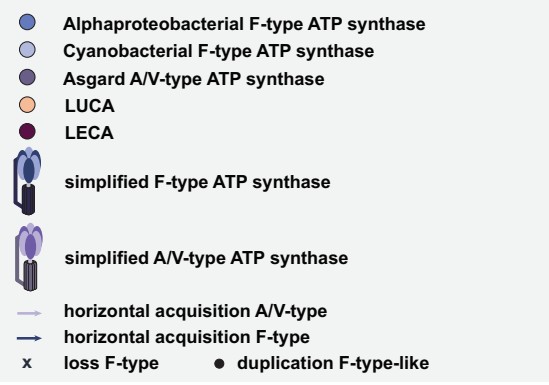

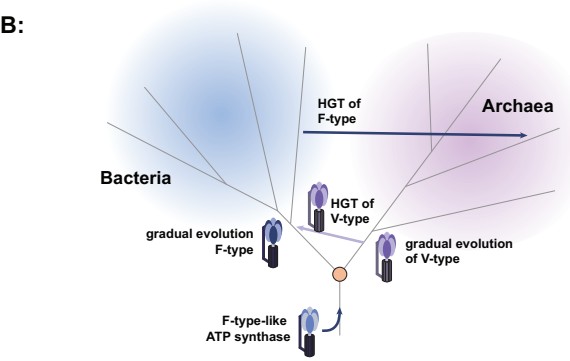

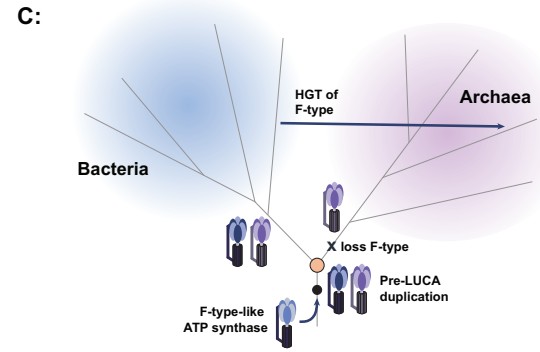

**Fig. 4 | ATP synthase evolutionary scenarios. A** Overview of possible ancestral ATP synthase acquisition in LECA from the putative prokaryotic hosts; the A/V-type derived from the archaeal host, and F-type ATP synthases derived from bacterial endosymbionts. **B** Evolutionary proposal supporting an F-type-like ancestral ATP synthase present pre-LUCA with subsequent divergence consistent with the split between Bacteria and Archaea and early transfers of A/V-type ATP synthases into the bacterial stem, and late HGT of F-type ATP synthases to Archaea. **C** Evolutionary proposal supporting an F-type-like ancestral ATP synthase and pre-LUCA duplication and divergence of at least the head components of the F- and A/V-type subunits with subsequent loss of the F-type components along the archaeal stem. The cartoon of the ATP synthase was drawn manually in Adobe illustrator.

The presence of all four subunits in Bacteria is consistent with ideas for a root of the universal tree within Bacteria[68–70]. However, we obtained significantly lower gene family likelihoods (approximately unbiased (AU) test): C60 model, pAU = 0.00009; C20 model, pAU = 0.0002) (Supplementary Data 6) for ATP synthase subunits reconciled with a species tree rooted within Bacteria rather than between Archaea and Bacteria[20,21]. When eukaryotes were excluded, the within-Bacteria root also had a lower likelihood, though not significant (C60, pAU = 0.093; C20, pAU = 0.547) (Supplementary Data 6). These results agree with the consensus root between Archaea and Bacteria.

To investigate the key motifs characterizing the catalytic and non-catalytic subunits of the F- and A/V-type ATP synthases we examined conserved protein motifs in extant taxa. We focused on the sequence identity of the Walker-A motif (Supplementary Discussion), which has an amino acid composition of *GXXXXGKT*[43]. This region comprises the primary "P-loop" domain responsible for binding phosphate during ATP synthesis/hydrolysis and is highly conserved across phosphate-binding proteins and fundamental to the activity of the ATP synthase[71,72]. Our analyses of the Walker-A motifs across the *nc*F1, *c*F1, and *c*A1V1 subunits revealed a conserved motif with variation in positions 2–5 (Fig. 3B). However, the *nc*A1V1 subunit lacks a recognizable Walker-A motif and instead contains a *SGSGLPHN* motif in the corresponding position (Fig. 3B). The phosphate binding properties of this motif are unknown[73].

We performed ancestral sequence reconstruction[46,47] on the alignment of the unrooted combined phylogeny (Fig. 3A, Supplementary Data 7) to determine the ancestral sequence at the root of each of the four subunits (*nc*F1: Node123; *c*F1: Node126; *c*A1V1: Node516; and *nc*A1V1: Node883) as well as the root of the catalytic versus non-catalytic subunits (Fig. 3A) (*nc*F1 and *nc*A1V1: Node124; *c*F1 and *c*A1V1: Node125). Consistent with our observations of the conserved extant motifs, we found Walker-A motifs in the reconstructed sequences for the *nc*F1, *c*F1, and *c*A1V1 families and the alternative motif (*SGSGLPHN*) for the *nc*A1V1 family (Fig. 3A). The alanine (A) and phenylalanine (F) dichotomy in the third position of the *nc*F1, *c*F1 and *c*AV ancestors is consistent with previous findings distinguishing F- and A/V-type ATP synthase catalytic binding loops, respectively (Fig. 3A)[41]. A motif pattern of *GGAGTGKT* was inferred for both the ancestor of the catalytic and non-catalytic subunits (Fig. 3A). This is compatible with a previously proposed scenario in which the progenitor ATP synthase was suggested to have contained six catalytic sites similar to the *c*F1[34]. While ancestral sequences inferred for the *nc*F1, *nc*A1V1, *c*F1 and *c*A1V1 are most similar to those in extant representatives of each of those families, the sequences inferred for each ancestor of the *nc* and *c* families were both most similar to extant members of the *c*F1 subunits from F-type ATP synthases (Fig. 3, Supplementary Data 7). Taken together, this may indicate that the ancestral head component of the ATP synthase was more similar to the R1 complex of F-type ATP synthase and is consistent with the hypothesis

that they evolved by duplication from a catalytic ancestor belonging to the "P-loop" NTPases[24,29,31,34,37]. Furthermore, our results imply that the ncA1V1 subunit lost its Walker-A motif after divergence from the other subunits, though the functional consequence of the degenerated binding loop in the ncA1V1 subunit is unknown.

## The origins of ATP synthases in eukaryotes

In agreement with symbiogenetic models for the origin of the eukaryotic cell[9–15], our ATP synthase phylogenies suggest that eukaryotes inherited A/V- and F-type ATP synthases from their archaeal and bacterial ancestors (Fig. 3A, Supplementary Figs. 4–10)[11,32–34,40]. Specifically, the relationship between Asgard archaea and eukaryotes was evident in phylogenies of the *nc*A1V1 subunit (Supplementary Figs. 6–10, Supplementary Data 8), with the strongest bootstrap support being 95.8/95 (Supplementary Fig. 7, Supplementary Data 8). In phylogenies for the catalytic subunits, the position of the eukaryotic branch was mostly unresolved (Supplementary Discussion, Supplementary Figs. 6–10, and Supplementary Data 8). This might be due to selective constraints or functional divergence considering that the eukaryotic V-type ATP synthase has evolved to couple proton transport to ATP hydrolysis rather than functioning as ATP synthase[32,34,42]. The origin of eukaryotic F-type sequences from Alphaproteobacteria and Cyanobacteria was consistently recovered across a range of analyses including both Bayesian and maximum-likelihood inferences (Supplementary Discussion, Supplementary Figs. 4–5, Supplementary Fig. 11, Supplementary Data 8, Zenodo data repository: https://doi.org/10.5281/zenodo.10012837[74]). Within the F-type subunits, the *nc*F1 phylogenies placed the sequences of eukaryotic plastids sister to *Gloeomargarita lithophora*, the closest living relative of the plastid[75], while the *c*F1 phylogeny grouped plastids together with most Cyanobacteria (Supplementary Fig. 11).

## Dating the species tree and establishing an absolute timeline for ATP synthase evolution

To establish a timeframe for the evolution of the ATP synthase, we built on the approach of Shih and Matzke (2013)[24] by bracing equivalent speciation nodes in both the ATP synthase phylogeny and a universal species tree. We took advantage of the greatly expanded sampling of organisms sequenced since the previous study (1520 total *nc* and *c* ATP synthase subunit sequences included in this study versus 149 total sequences[24]) and applied more fossil calibrations (ATP synthase gene tree $n = 10$, species tree $n = 12$ versus $n = 7$[24]) (Supplementary Discussion, Supplementary Data 9). We developed a new molecular dating software (McmcDate) that implements both cross-bracing (two nodes constrained to the same age) as well as relative age constraints (one node is constrained to always be younger than another node, as informed, for instance, based on horizontal gene transfer between donor and recipient lineages[49,76]) (Supplementary Discussion, Supplementary Data 9).

Molecular dating analyses revealed that bracing the nuclear, mitochondrial, and plastid eukaryotic clades had a significant overall impact (*Z*-test statistic was −233.0 with a *p*-value of 0.0) on inferred rates of evolution, which are 16.5% higher overall than in the non-braced analysis (2.6e-4 average number of substitutions per million years and site, for ribosomal protein tree; Fig. 5A, C, Supplementary Figs. 12–17, Supplementary Data 10). As a result, age ranges (measured as 95% highest posterior density: the boundaries of the central 95% highest posterior densities of the distributions on ages) are modestly younger in the braced analysis, though similar overall (Supplementary Data 10). We estimate that LUCA lived 4.52–4.32 Ga and 4.52–4.42 Ga in the braced and non-braced analysis, respectively (Fig. 5A, Supplementary Fig. 16). Ages towards the younger end of the spectrum from our braced analysis seem more plausible considering the Moon-forming impact at 4.52 Ga, though both ages imply a rapid origin of LUCA following this putative sterilization event. In the following, we

focus on dates from the cross-braced analysis but corresponding age ranges without bracing can be found in Fig. 5 and Supplementary Data 10). Of the two prokaryotic domains, LBCA was inferred to be older than LACA (4.49–4.05 Ga versus 3.95–3.37 Ga) indicating higher extinction or lower sampling rates for the archaeal stem-lineage (Fig. 5A, Supplementary Fig. 16). Our analyses suggest that eukaryotes diverged from their closest known asgardarchaeal relatives 2.67–2.19 Ga (Hodarchaeota + eukaryotes), and from Alphaproteobacteria 2.58–2.12 Ga (Supplementary Fig. 16, Supplementary Data 10). Plastids diverged from free-living Cyanobacteria 2.14–1.73 Ga (Fig. 5A, C, Supplementary Fig. 16, Supplementary Data 10) and we inferred LECA to have originated 1.93–1.84Ga (Supplementary Fig. 16, Supplementary Data 10). These revised ages for key nodes in the species tree provide a timeline to study ATP synthase diversification in the context of cellular evolution: the split between the catalytic and non-catalytic ATP synthase subunits (4.52–4.46 Ga) likely predates (or at the latest was contemporary with) LUCA (4.52–4.32 Ga), while the divergence into F1- and A1/V1-types within the catalytic (4.52–4.38 Ga) and non-catalytic (4.52–4.42 Ga) clades overlaps in time with LUCA (4.52–4.32 Ga) and LBCA (4.49–4.05 Ga) but predates LACA (3.95–3.37 Ga) by more than 0.5 Gyr (Fig. 5B, Supplementary Figs. 18–19, Supplementary Data 10). An early origin of the A/V-type ATP synthases is a prerequisite for their presence in LBCA. If the split between F- and A/V-types corresponds to the speciation of Archaea and Bacteria, an older age for the A1/V1 clade compared to crown Archaea (LACA) might hint at a sampling or extinction "bottleneck" on the stem lineage leading to extant Archaea (Fig. 4B).

## Discussion

The results of our analyses confirm that the A/V-type ATP synthase was present in LACA[67] and the F-type ATP synthase was in LBCA[19–22]. They also revealed that A/V-type ATP synthases are broadly distributed in Bacteria and might have already been present in LBCA (Fig. 1, 3A, Supplementary Fig. 2, Supplementary Figs. 6–8, Supplementary Fig. 10). Previous analyses suggested that the acquisition of A/V-type ATP synthases in Bacteria occurred via HGT from hyperthermophilic Archaea[33,35,53], despite the observation that many mesophilic Bacteria also contain A/V-type ATP synthases (Figs. 1, 3A, Supplementary Fig. 2). In contrast, only three archaeal genomes (all within the genus *Methanosarcina*) appear to encode F-type ATP synthases (Figs. 1, 3A, Supplementary Fig. 2, Supplementary Figs. 4–5) belonging to a family of ATP synthases known as N-ATPases. The latter represent a distinct horizontally-acquired F-type ATP synthase which exists in addition to a bona fide F- or A/V-type ATP synthase in Bacteria and Archaea[56]. Experimental studies revealed that the N-ATPase of *M. acetivorans* is not required for growth[55,56] and the function is debated.

ATP synthase evolution in Bacteria seems to be driven by frequent transfers from Archaea to Bacteria. Alternatively, A/V-type ATP synthases may already have been present in LBCA or LUCA and subsequently lost in many bacterial lineages. The latter possibility is in line with a scenario in which transfers from Bacteria to Archaea have been more common during evolution[77], but requires a loss of the F-type ATP synthase along the branch leading to Archaea (Fig. 4C). In agreement with this, the duplication giving rise to the catalytic and non-catalytic subunits (4.52–4.42 Ga 95% highest posterior density, Supplementary Figs. 18–19, Supplementary Data 10) and the divergence into F- and A/V-lineages within the catalytic and non-catalytic clades (4.52–4.38 Ga 95% highest posterior density, Supplementary Figs. 18–19, Supplementary Data 10) were inferred to have occurred very early in the history of cellular life, prior to, or overlapping with, LUCA (4.52–4.32 Ga, Fig. 5A, C, Supplementary Fig. 16, Supplementary Data 10). This may seem at odds with previous inferences[10,29,32,78] and suggestions of deep divergences between Archaea and Bacteria coinciding with distinct informational processing machinery[19], ATP synthases, and membrane lipids[36,79]. However, the 'lipid divide'[80] appears less pronounced than assumed

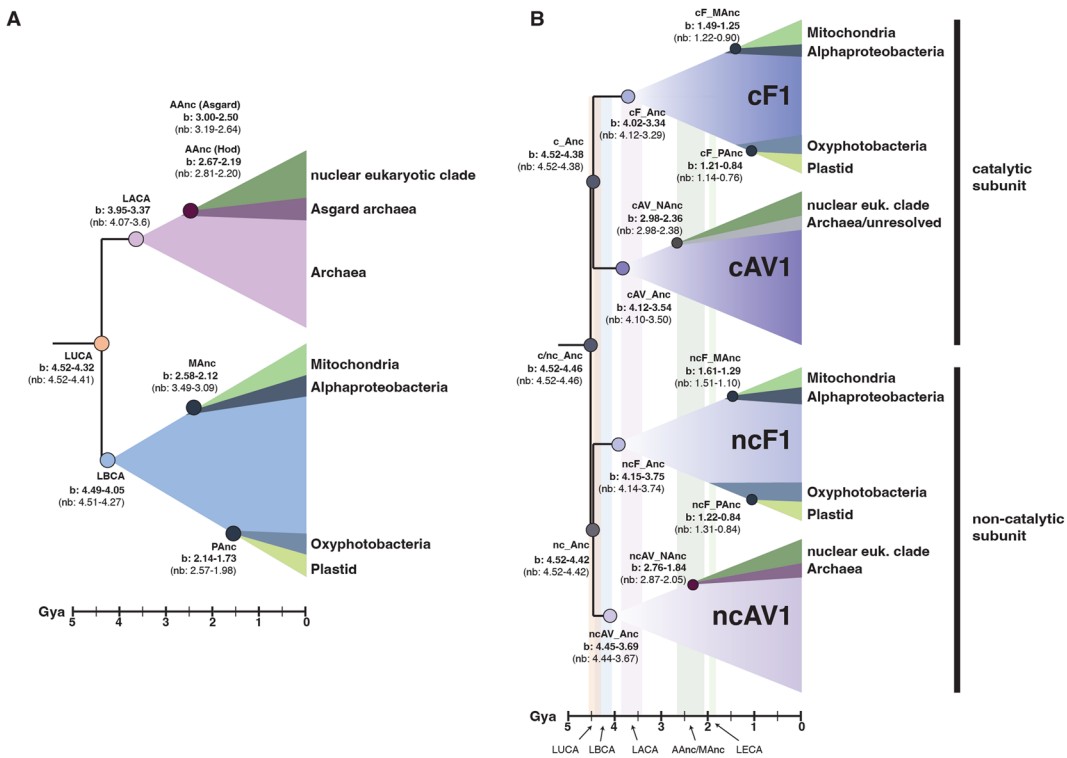

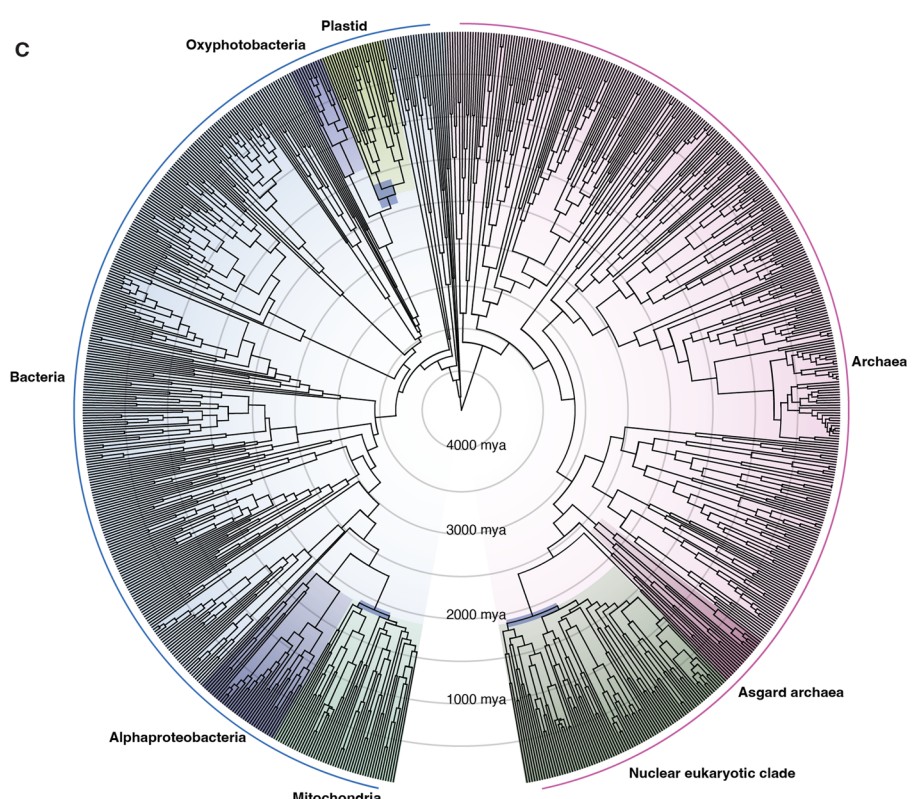

previously[81,82] and LUCA may have had the mevalonate pathway[83,84] and been able to synthesize bacterial and archaeal-type lipids[21,85]. The occurrence of the F- and/or A/V-type ATP synthases in modern Bacteria, suggests that their membrane lipids are compatible with either ATP synthase type. It is unclear whether the lack of Archaea with a complete substitution of an A/V- with F-type ATP synthase can be explained by constraints imposed by archaeal membrane lipid composition.

Alternatively, it is possible that while the diversification of the catalytic and non-catalytic subunits into the respective F1 and A1/V1 families may have predated LUCA, the hexameric headpieces may have functioned independently from extant membrane components[29]. In this scenario, the evolution of the membrane components could have occurred later, potentially in conjunction with the speciation of Bacteria and Archaea and the 'lipid divide'[36,79].

**Fig. 5 | Timing of cellular evolution across the tree of life based on a cross-braced dated ribosomal species tree and ATP synthase gene tree. A** Suggested timing of key evolutionary events based on a schematic ribosomal species tree. **B** Suggested timing of key evolutionary events based on a schematic ATP synthase gene tree. **C** Dated cross-braced ribosomal species tree (Edited2, see Methods) including nuclear, mitochondrial, and plastid eukaryotic homologs. See Methods for inference of the maximum-likelihood concatenated ribosomal species phylogeny and constraints (Edited2). The alignment contained 863 sequences and was

trimmed with TRIMAL[129] (alignment length = 2133 amino acids), and the maximum-likelihood phylogeny was inferred using IQ-TREE2 v2.1.2 the LG+C60+R+F model[111,122,123]. Abbreviations: b braced, nb non-braced, MAncL shared ancestor of mitochondria and closest alphaproteobacterial sister lineage, AAnc: shared ancestor of eukaryotic host lineage and closest asgardarchaeal sister lineage; Hod, Hodarchaeales; PAnc, shared ancestor of plastid and closest cyanobacterial sister lineage; c catalytic; nc non-catalytic; F F-type ATP-synthase; AV AV-type ATP-synthase.

Despite the wide distribution of ATP synthases across cellular life, our analyses revealed that many DPANN Archaea and CPR Bacteria may have minimal complexes, as is the case in the DPANN *Nanoarchaeum equitans*[86], or even lack all genes for an ATP synthase complex (Fig. 1, Supplementary Fig. 2, Supplementary Data 4). This finding suggests that ATP synthases are not as essential as previously assumed[32,33,45,65] with loss in DPANN and CPR lineages likely being the result of genome streamlining processes consistent with their predicted host-dependent lifestyles[58,59]. For instance, various members of DPANN lack several ATP synthase subunit homologs (60/103) while others encode homologs clustering with other DPANN or potential hosts (Fig. 1, Supplementary Figs. 6–10, Supplementary Data 4). We observed putative symbiont-host gene transfers between acidophilic Micrarchaeota and their hosts belonging to the Thermoplasmata[87] consistent with work supporting extensive HGT among ATP synthase genes of acidophilic archaeal lineages[60] (Supplementary Figs. 6–10, Supplementary Data 4). Furthermore, our trees indicate HGT and/or compositional attraction[88] of the *c*A1V1 subunit between the symbiotic Nanohaloarchaeota and their halobacterial hosts (Supplementary Fig. 6). The evolution of ATP synthase genes underpins debate over the phylogenetic placement of Nanohaloarchaeota[45], originally placed as the sister-group to Halobacteria[45,89] but later recovered as a member of DPANN[16,17,90–92]. Recently, Feng and coworkers found that the catalytic and non-catalytic subunits of the A/V-type ATP synthase of Nanohaloarchaeota form sister-groups to halobacterial homologs[45]. Although their concatenated species trees placed Nanohaloarchaeota with other DPANN[45], the authors argued that this placement was an artifact due to compositional biases in the concatenated dataset, with the ATP synthase gene tree recording the true organismal history. By contrast, Wang et al. (2019)[60] suggested that the incongruence of the species and ATP synthase gene trees for halophilic Archaea result from the HGT of an ATP synthase operon from Halobacteria into the common ancestor of Nanohaloarchaeota. Ecological association and symbiotic interactions between these organisms might have facilitated such a transfer. We include a larger representation of DPANN and metagenome-assembled genomes (GCA_003660905, GCA_003660865) belonging to a divergent sister lineage of the Nanohaloarchaeota (Fig. 1, Supplementary Fig. 2, 20), providing an opportunity to reconsider and distinguish hypotheses. Our results group Nanohaloarchaeota *nc*A1V1 subunits with Halobacteria, while in the *c*A1V1 subtree, Nanohaloarchaeota group with DPANN (including the sister lineage, Supplementary Figs. 6–10). Our analyses are most compatible with a scenario in which the last common ancestor of Nanohaloarchaeal already possessed an ATP synthase complex inherited vertically from its DPANN relatives. The *nc*A1V1 subunit may have been replaced through HGT from a halobacterial host early during the lineage's evolution, potentially as an adaptation to halophily. Alternatively, it might be compositionally attracted to homologs of the Halobacteria as a result of convergent adaptations to halophily[88]. This suggests that even genes whose synteny is conserved across lineages may be individually affected by HGT or evolutionary constraints. In such cases, the phylogenetic signal encoded in a larger number of marker genes may provide a more reliable estimate of the species tree.

Consistent with observations that niche expansion of Thaumarchaeota into acidic soils and high pressure oceanic zones was linked to their horizontal acquisition of a variant V-type ATP synthase

operon[60], our results illustrate the potential role of symbiont-host gene exchange and environmental factors in ATP synthase evolution. Prospective studies focusing on genome evolution of DPANN archaea can help further assess the presence of ATP synthases and other metabolic components in the various DPANN ancestors and elucidate instances of transfer and loss of genes throughout DPANN diversification and adaptation to their respective symbiotic hosts.

Our molecular clock analyses suggest cross-bracing species nodes within gene trees is effective in propagating temporal information across the tree of life and improves the precision and accuracy of divergence time estimates for Archaea and Bacteria. Bracing resulted in higher estimated rates of molecular evolution overall (Supplementary Fig. 17, Supplementary Data 10), with the result that various deeper nodes of the tree were estimated to be slightly younger when compared with the un-braced analyses. This also includes LUCA, which has a mean age of 4.46 Ga in our cross-braced analyses and of 4.49 Ga in un-braced analyses. However, as observed in previous studies[19,27], the credibility interval associated with the LUCA still clashes against the root hard-maximum represented by the moon-forming impact even when implementing cross-bracing. This indicates that bracing helps to ameliorate, though not completely resolve, the problem of an under-calibrated clock inferring rates that are too low to account for the amount of genetic change that has occurred since the root of the universal tree[19,93]. Some recent studies have reported moderately younger age estimates for LUCA: 4.05–3.42 Ga[94], or a range of values 4.48–3.93 Ga depending on conditions[95]. An important driver of these differences is the choice of root maximum, which was younger in both studies (3.8–4.1 Ga[95] and 3.9 Ga[94]). In turn, also in those studies[94,95], the credibility interval for the age of LUCA clashes against the maximum used to calibrate the root node. This is consistent with previous work suggesting that the age of LUCA is sensitive to the root calibration used[19,21,27,95,96]. We used the age of the Earth as our root maximum (the moon-forming impact at 4.52 Ga) because we are unaware of any compelling evidence for a younger maximum on the age of extant life (Supplementary Material). Thus, while the precise age of life and of LUCA remains uncertain, the inferred ages of LUCA and the early ATP synthase duplicates seem to imply a very high rate of evolutionary innovation during the earliest period of evolutionary history. Additional calibrations for deep nodes in the universal tree, along with date estimates for other pre-LUCA paralogs, may help to dissect this key evolutionary period in higher resolution in future work (see Supplementary Discussion for further details about the resulting age estimates for major prokaryotic clades).

The LECA estimate (1.93–1.84 Ga, 95% highest posterior density) from our species tree analysis falls within published molecular clock estimates, placing LECA within a broad interval ~1–2.4 Ga[25–27,97–99] (Fig. 5A, C, Supplementary Fig. 16, Supplementary Data 10). More recent analyses have tended to resolve an older LECA, with ages closer to 1 Ga being less plausible on the basis of fossils from that period that can uncontroversially be assigned to crown Archaeplastida. These fossils include the green alga *Proterocladus antiquus* (1 Ga)[100] and the red alga *Bangiomorpha pubescens* (>1030 Mya)[101]. The ages of some of these nodes, including LECA and particularly the last plastid common ancestor (LPCA), were inferred to be younger in the ATP synthase analysis (Fig. 5, Supplementary Figs. 16 and 18, Supplementary Data 10). In part, this may be due to the shorter alignment of ATP

synthase (433-512AA, Supplementary Figs. 4–7, 10) and lower phylogenetic resolution[102] reducing the species tree calibrations and braces to the ATP synthase phylogeny through gene and species tree incongruence (Supplementary Information).

Our analyses are of interest for the timing of mitochondrial acquisition relative to other hallmark features of eukaryotes such as the nucleus[103,104] and help to explain the differences in the length of the stem between eukaryotic genes of archaeal and bacterial origin reported previously[103,105]. Note that while the LECA nodes within the mitochondrial and nuclear lineages can be cross-braced to be contemporaneous, the lengths of the antecedent stems (i.e. the divergence times of the mitochondrial and nuclear lineages from their closest bacterial and archaeal relatives) might be very different (there should be no expectation that they are of equal antiquity). Our analyses support a moderately longer stem for the nuclear lineage (mean: 520.3 Ma, 291–789 Ma, 95% highest posterior density) than the mitochondrion (mean: 438.8 Ma, 233–682 Ma, 95% highest posterior density), suggesting the divergence of the nuclear lineage from the closest sampled Asgard archaea occurred before the divergence of the mitochondrial lineage from Alphaproteobacteria. However, the credible age ranges for these divergences overlap, therefore some additional factor (e.g. a faster evolutionary rate prior to LECA in eukaryotic genes of archaeal origin) may contribute to the observed differences in stem lengths[103,105]. Interestingly, the inferred timescale is sensitive to the phylogenetic position of eukaryotes within Asgard archaea and Alphaproteobacteria: in an alternative analysis in which eukaryotes were placed sister to all Asgard archaea, and mitochondria within Alphaproteobacteria, the difference in stem group ages was more pronounced (mean 812.1 Ma, 95% highest posterior density 540–1105 Ma, nuclear stem: 310.1 Ma, mitochondrial stem: 150–508 Ma) (Supplementary Fig. 12). While this result tells us something about the shape of the tree of life it does not distinguish between hypotheses of an "early" or "late" mitochondrial acquisition. This is because these hypotheses make competing predictions about the order in which key eukaryotic features without direct correspondence to nodes in the tree were acquired relative to the mitochondrial endosymbiosis (Donoghue et al. 2023)[106].

## Concluding remarks

Our analyses provide insights into the diversification of the ATP synthase gene family and established age estimates for key nodes in the tree of life. Our results suggest that while LACA solely harbored an A1/V1-type ATP synthase, LBCA may already have encoded homologs of the head component of both the F- and a A/V-type ATP synthase. Studying how A/V-type ATP synthases function in Bacteria will help to explain the distribution we observed and the functional consequences of the ancient divergence between F- and A/V-type ATP synthases. In contrast to previous work, our inferences are consistent with the hypothesis that the divergence of the F1- and A/V1-type ATP synthase components may have predated LUCA. Furthermore, ATP synthase evolution supports scenarios on eukaryotic origins from an asgardarchaeal host[3,4,13,14] and alphaproteobacterial symbiont[107,108] and, together with our dated species tree, provide an updated timescale of cellular evolution, placing the origin of the eukaryotic cell into a geological context that can help to test eukaryogenesis models.

## Methods

### Selection of 800 taxa comprising the backbone genome reference dataset

**Archaeal reference genomes.** A representative set of archaeal genomes was selected from a broad diversity of all archaeal genomes present in NCBI. A set of 51 marker proteins[91] was used to infer an initial concatenated phylogeny of 574 archaeal genomes meeting a threshold of >40% completeness and <13% contamination (Supplementary Data 1). Individual markers were aligned with MAFFT L-INS-i

v7.407 (settings: −reorder)[109], trimmed using BMGE v1.12 (settings: -m BLOSUM30 -h 0.55)[110] and concatenated with a custom script (catfasta2phyml.pl; https://github.com/nylander/catfasta2phyml). A phylogenetic tree was generated with IQ-TREE v1.6.7 (settings: -m LG +C60+F+R -bb 1000 -alrt 1000)[111].

Based on this tree, 350 archaeal genomes were subsampled to evenly represent archaeal phylogenetic diversity (Supplementary Data 1). Type-strains were preferentially selected, while high quality metagenome assembled genome and single cell assembled genomes were selected based on completeness and contamination levels.

**Bacterial reference genomes.** The bacterial reference backbone, prioritizing type-strains and reference genomes, but also high-quality metagenome assembled genomes and a subselection of representatives from candidate phyla, was derived using an initial phylogeny of bacterial genomes available in NCBI as described above. Homologs of a conserved set of 29 marker proteins, i.e. a subset of 48 single-copy marker proteins previously defined in Zaremba-Niedzwiedzka et al. (2017)[4] were identified in those bacterial genomes, aligned using MAFFT v7.407 (settings: −reorder)[109], trimmed using BMGE v1.12 (settings: -m BLOSUM30 -h 0.55)[110] and concatenated to reconstruct a phylogenetic tree using IQ-TREE v1.6.7 (settings: -m LG+G -bb 1000 -alrt 1000)[111]. We subsampled the concatenated phylogeny for 349 bacterial genomes that represent known bacterial genomic diversity, ensuring selection of major bacterial taxonomic clades. The genome of *Schaalia odontolytica* ATCC 17982, which represents the host of members of the Saccharibacteria (formerly phylum TM7)[112,113], was downloaded from NCBI in 2020 and manually added to the bacterial backbone dataset (Supplementary Data 1).

**Eukaryote reference genomes.** A set of 100 published genome-wide datasets (genomes and, for lineages lacking complete genomes, largely complete transcriptomes) were sampled to represent the major lineages of eukaryotes (Supplementary Data 1, Supplementary Data 11). We also included sequences from the unpublished *Diplonema papillatum* genome project, with the permission of the sequencing consortium (see Acknowledgements).

**Functional annotations.** To identify sequences of ATP synthase subunits within all genomes in the 800-backbone set, all protein coding sequences were annotated using the KEGG and COG databases. Sequences were compared to KO profiles within the KEGG Automatic Annotation Server (KAAS, downloaded April 2019) (KAAS; downloaded April 2019)[114], to COG profiles within the NCBI COG database (downloaded May 2020)[115–117], and to Pfam profiles in the Pfam database (Release 34.0)[118]. KOs and COGs were assigned using hmmsearch v3.1b2 (settings: −tblout sequence_results.txt -o results_all.txt −domtblout domain_results.txt −notextw -E 1e-5)[119]. Pfams were assigned using hmmsearch v3.1b2 (settings: --tblout sequence_results.txt -o results_all.txt --domtblout domain_results.txt --notextw -E 1e-10)[119].

### Inference of a concatenated species phylogeny including Archaea, Bacteria, and eukaryotes

**Marker gene homology search.** A concatenated phylogeny of the 800 bacterial, archaeal, and eukaryotic genomes included in this study was inferred using a previously defined set of 27 single-copy marker genes[19] (Supplementary Fig. 20, Supplementary Data 12). To collect the corresponding homologs, the 800 reference genomes were queried against all COG HMM profiles with a custom script built on the hmmsearch [options] <reference genomes> <hmmfile> algorithm[120]: hmmsearchTable Whole_ArcBacEuk_800_vs2_clean.faa NCBI_COGs_Oct2020.hmm -E 1e-5 > 1_Hmmersearch/HMMscan_Output_e5 (HMMER v3.3.2)[121], and all homologs corresponding to the 27 single-copy marker genes were identified, cleaned, and parsed. The

approaches used to identify the appropriate homologs for prokaryotes and eukaryotes are described below.

**Selection of prokaryotic homologs.** For prokaryotes, the best-hit sequences were selected based on e-value and bitscore and the corresponding protein sequences were extracted from the reference genome backbone. Protein sequences assigned to each marker gene were aligned using MAFFT L-INS-i v7.453 (settings: --reorder)[109] and trimmed using BMGE v1.12 (settings: -t AA -m BLOSUM30 -h 0.55)[110]. Maximum-likelihood phylogenies with ultrafast bootstrap approximation (UFBoot) for each single-copy marker gene were constructed using IQ-TREE2 v2.1.2 (settings: -m LG+G -wbtl -bb 1000 -bnni)[111,122,123]. Individual marker gene trees were manually inspected for domain-level monophyly, the presence of paralogous protein families, and signs of contamination including LBA and horizontal gene transfer (HGT) (Supplementary Data 1, Zenodo data repository: https://doi.org/10.5281/zenodo.10012837[74]). Marker genes, where domain-level lineages were paraphyletic were excluded and sequences with indications of LBA, HGT, and paralogy were manually removed using a custom script: remove_seq_with_specific_header3.py.

**Selection of nuclear eukaryotic homologs.** To distinguish between the nuclear, plastid, and mitochondrial homolog and select the correct eukaryotic representative sequence, we collected all eukaryotic hmmsearch hits and downsampled them with CD-HIT v4.7 using a threshold of 90% sequence identity[124,125]. The filtered eukaryotic sequences were combined with the previously inspected prokaryotic sequences and all sequences for each single-copy marker gene were aligned using MAFFT L-INS-i v7.453 (settings: --reorder)[109], and trimmed using BMGE v1.12 (settings: -t AA -m BLOSUM30 -h 0.55)[110]. Single gene phylogenies were inferred using FastTree (settings: -lg)[126]. KEGG and Pfam annotations (see above) were mapped to the tips of the eukaryotic sequences for manual inspection of multiple paralogs per taxon. First, the eukaryotic sequences were inspected by removing any sequence failing monophyly (i.e., HGTs in prokaryotic clades) or not clearly derived from the nuclear source (i.e., the plastid and/or mitochondrial sequences). Duplicate nuclear eukaryotic sequences were filtered in a two-step procedure: (1) if duplicate sequences are monophyletic, select a single representative based on protein annotation consistent with the identity of the single-copy marker gene, and (2) if duplicate sequences are paraphyletic, remove taxon completely from the single-copy marker gene. Any representatives with fewer than 65% of the marker genes (20 taxa removed, 80 eukaryotes in total) were removed from this analysis (Supplementary Data 13).

**Inspection of final marker gene sequence sets.** The final set of eukaryotic nuclear sequences were combined with the previously inspected sequences for Archaea and Bacteria (see above) and aligned using MAFFT L-INS-i v7.453[109], trimmed with BMGE v1.12 (settings: -t AA -m BLOSUM30 -h 0.55)[110], and single gene trees were inferred using maximum-likelihood with UFBoot approximation methods in IQ-TREE2 v2.1.2 (settings: -m LG+G -wbtl -bb 1000 -bnni)[111,122,123]. Upon inspection of single gene trees including homologs from Archaea, Bacteria and eukaryotes, six single-copy markers (COG0064, COG0085, COG0086, COG0202, COG0480, and COG5257 (Supplementary Data 12) were flagged for removal (e.g. lack of clear nuclear paralog, or absence of archaeal or bacterial sequences in the tree).

**Inference of the concatenated phylogeny.** Alignments for the 21 single-copy marker genes were generated and trimmed following the approaches outlined above and individual marker alignments were concatenated using the script catfasta2phyml.pl (https://github.com/nylander/catfasta2phyml). The final concatenated sequence alignment contained 3367 positions and was used to infer maximum-likelihood phylogenies using varying models of evolution in IQ-TREE2 v2.1.2

(settings: -m LG+C60+R+F *or* LG+C20+R+F -bb 1000 -alrt 1000)[111,122,123]. We examined the statistical support for topologies of the two concatenated species trees inferred under different models of evolution (LG+C60+R+F and LG+C20+R+F, see above, Supplementary Data 14, Zenodo data repository: https://doi.org/10.5281/zenodo.10012837[74]) using the approximately unbiased (AU) test implemented in IQ-TREE2 v2.1.2 (settings: -s 21eLife_ArcBacEuk_wHuber_vs1.faa -m [LG+C20+R +F/LG+C60+R+F] -z 21eLife_ArcBacEuk_wHuber_vs1_bothtrees.treefile -pre [C20/C60] -n 0 -zb 10000 -au[111,123,127]. Results are shown in Supplementary Data 15. Despite statistical exclusion of the LG+C20+R+F topology, we chose to use this tree for phylogenetic interpretation because the placement of key lineages such as the Asgard archaea and CPR, is most consistent with recent evidence[4,19,21,22] (Supplementary Fig. 20, Supplementary Data 15).

## Constructing a ribosomal marker phylogeny including nuclear, mitochondria, and plastid homologs

**Selection of eukaryotic nuclear, mitochondrial, and plastid homologs.** Eukaryotes encode two or more ribosomes of distinct prokaryotic origins, i.e., archaeal, alphaproteobacterial and, in the case of the presence of a plastid, a cyanobacterial origin (i.e. the nuclear, mitochondrial, and plastid, respectively). A concatenated phylogeny including, if identified, the nuclear, mitochondrial, and plastid ribosomal protein homologs for each eukaryote, was inferred for molecular dating and bracing analyses. To this end, we constructed single gene trees of the 15 ribosomal marker genes (subset of the 21 single-copy marker genes described above) which included Archaea, Bacteria, and all eukaryotic homologs (i.e., the nuclear, mitochondrial, and plastid). Note that the nuclear eukaryotic sequences were the same set of sequences reported in the final inspection of the concatenated species phylogeny (see above). To identify the plastid homologs, we selected the monophyletic clade of eukaryotic sequences affiliated with the Cyanobacteria. The mitochondrial sequences appeared to demonstrate variable placements with some affiliating with the alphaproteobacteria and others branching basally in the Bacteria. Therefore, we made our sequence selection based on the position of known mitochondrial genes of the type-species *Homo sapiens* and *Saccharomyces cerevisiae*. First, we manually located *H. sapiens* in the phylogenies and searched the protein accession in Uniprot and/or NCBI[128] to confirm sequence annotation and identity as a mitochondrial sequence. In the absence of a *H. sapiens* homolog, we used *S. cerevisiae* mitochondrial homologs. Of the 15 ribosomal markers, three had no distinguishable mitochondrial homolog for either type-species and were dropped from the dataset, resulting in 12 ribosomal markers (Supplementary Data 16). All eukaryotic sequences, that clustered with the *H. sapiens*/*S. cerevisiae* homolog and grouped with alphaproteobacteria or basally in the phylogeny, were selected for subsequent analyses. Selected sequences were de-replicated using the following criteria: (1) if paralogous sequences are monophyletic retain one homolog based on annotation or manual selection, and (2) if paralogous sequences are paraphyletic remove all sequences from that organism. Dereplicate sequences marked for removal are in Supplementary Data 16. Gene trees with selected sequences have been deposited in our data repository at Zenodo: https://doi.org/10.5281/zenodo.10012837[74].

Ribosomal protein homologs were then annotated based on their distinct origin (nuclear, mitochondrial, plastid) and the percent distribution of homologs across the 12 ribosomal markers by eukaryotic taxon was calculated. Only taxa that had at least 50% of the markers of nuclear, mitochondrial, or plastid origin were retained, resulting in 88 nuclear taxa, 50 mitochondrial taxa, and 25 plastid taxa (Supplementary Data 13). The eukaryotic GenomeIDs for each sequence were annotated with the suffix of the origin (i.e., EukGenome_nuclear, EukGenome_mito, etc.) for downstream concatenation. In total, the sequence sets for the 12 ribosomal markers contained archaeal and

bacterial homologs, and the eukaryotic nuclear, mitochondrial, and plastid sequences, respectively. Alignments were generated using MAFFT L-INS-i v7.453 (settings: --reorder) [109] and trimmed with TRIMAL v1.2rev59 (settings: -gappyout) [129]. The alignments of the 12 ribosomal markers were concatenated using the script catfasta2phyml.pl (https://github.com/nylander/catfasta2phyml) and the final concatenated alignment contained 2133 sites.

**Inference of concatenated phylogenies.** A maximum-likelihood phylogeny was inferred using IQ-TREE2 v2.1.2 (settings: -m LG +C60+R+F -bb 1000 -alrt 1000) [111,122,123] (Supplementary Fig. 21).

**Assessing distribution of ATP synthase genes across 800 taxa backbone**
We performed a comparative genomic analysis of the distribution of ATP synthase genes across the 800 taxa included in this study. COG families corresponding to each subunit of the ATP synthase (Supplementary Data 3) were extracted from the 800 reference genomes. Results were compiled, counted in R v4.1.1 (Supplementary Data 4). The count table was converted to a binary presence/absence matrix that was summarized using the ddply function of the plyr package (v1.8.6) by the respective phylogenetic clustering methods: (1) species-level according to order of individual species in the inferred concatenated phylogeny (BinID and Tip_Order, Supplementary Data 17), and (2) class- and phylum-level for Archaea and phylum-level for Bacteria corresponding to clade clustering in the concatenated phylogeny (CladeCluster and Clade_Order, Supplementary Data 17). The percentage distribution of subunits within each phylogenetic cluster was visualized in a bubble plot implemented using the ggplot function with geom_tile and facet_grid from the ggplot2 package v3.3.5. The binary presence/absence of subunits by species was visualized with the ggplot function using geom_point and facet_grid from ggplot2 v3.3.5. All heatmaps and bubble plots were manually merged with the corresponding concatenated species tree in Adobe Illustrator CC v22.0.1.

**Curation of eukaryotic hits.** We conducted an additional step of quality control to curate eukaryotic protein sequences potentially corresponding to the key ATP synthase subunits highlighted in Supplementary Data 3. All eukaryotic proteins suggested to represent homologs of ATP synthase COGs (Supplementary Data 3) were identified in the protein annotation table (Supplementary Data 2) and the corresponding sequences were queried against the NCBI non-redundant (NCBI_nr) database using diamond blast v2.0.8 (settings: diamond blastp -q ${sample}_seqs.faa --more-sensitive --evalue 1e-5 --threads 20 --seq 100 --no-self-hits --db nr.dmnd --taxonmap prot.accession2taxid.gz --outfmt 6 qseqid qtitle qlen sseqid salltitles slen qstart qend sstart send evalue bitscore length pident staxids) [130]. We ranked the hits by e-value and bitscore and collected the (up to) top 10 hits per accession. Taxonomic information was mapped to the table using the NCBI taxonomy corresponding to the taxid. Domain identity for the top 10 hits per protein sequence were summarized and any sequence with ≥50% hits to Bacteria was considered putative bacterial contamination and flagged for removal (Supplementary Data 4). In total, 326 accessions were removed and not considered for the presence-absence analysis of the ATP synthase subunits (Supplementary Data 4). Putative contamination was also inspected in the protein sequences used to infer ATP synthase gene phylogenies and four sequences have been highlighted as putative bacterial contamination (Supplementary Data 4, Supplementary Figs. 4–5, Zenodo data repository: https://doi.org/10.5281/zenodo.10012837 [74]).

**Phylogenetics of ATP synthase subunits**
**Sequence retrieval and selection.** Interpro domains that characterize the protein families corresponding to the subunits present in the

catalytic (R1) domain of the F-Type and A/V-Type ATP synthases were selected at the family-level [131] and include: ipr005294 (F-Type alpha, hereafter *nc*F1), ipr005722 (F-Type beta, hereafter *c*F1), ipr022878 (A/V-Type A, hereafter *c*A1/V1), and ipr022879 (A/V-type B, hereafter *nc*A1/V1). All protein sequences assigned to the corresponding interpro domains were extracted from the UniProt Knowledge Base [128], and were searched against the 800 reference genomes using DIAMOND v0.9.22.123 (settings: blastp -p 4 -f 6 qseqid stitle pident length mismatch gapopen qstart qend sstart send e-value bitscore) [130]. Top hits were selected based on best e-value and sequence identity, and all unique protein accessions (from the 800 reference taxa) were used to extract the amino acid sequences from the 800-genome reference dataset. Sequences with undefined characters (i.e., X, x) and/or outside of the average sequence length of homologs, i.e., 300–675 bp, were filtered from the sequence sets. To avoid highly similar duplicates, sequences with 99–100% identity were removed using CD-HIT [124,125]. Additionally, for consistency with the concatenated species phylogeny (see above), eukaryotic taxa that fell below the 65% marker gene presence cutoff (20 eukaryotic taxa, Supplementary Data 13) were removed from the single-subunit sequence sets.

**ATP synthase subunit phylogenies: *c*F1, *nc*F1, *c*A1V1, *nc*A1V1**
A series of seven different sequence sets were generated for analysis:
1. Single subunits sets F1-alpha (*nc*F1), F1-beta (*c*F1), A1/V1A (*c*A1V1), and A1/V1B (*nc*A1V1) (four in total)
2. Combined orthologous subunits for outgroup rooting: F1A+A1/V1B (*nc*F1+*nc*A1V) and F1B+A1/V1A (*c*F1+*c*A1V1)
3. All four subunits combined

Potential duplicates were removed from the combined sets using CD-HIT v4.7 with a 100% identity (settings: cd-hit -1) [124,125], sequences were aligned using MAFFT L-INS-i v7.453 (settings: --reorder) [109] and trimmed using BMGE v1.12 (settings: -m BLOSUM30 -h 0.55) [110]. The best-fit model was determined using the Model Finder Plus tool implemented in IQ-TREE2 v.2.1.2 (settings: -m MFP -mset LG -madd LG +C10,LG+C20,LG+C30,LG+C40,LG+C50,LG+C60,LG+C10+R+F,LG +C20+R+F,LG+C30+R+F,LG+C40+R+F,LG+C50+R+F,LG+C60+R+F --score-diff all -bb 1000 -alrt 1000 -bnni -wbtl) [111,122,123,132] and the best-fitting model for each gene tree was selected based on the Bayesian Information Criterion (BIC, Supplementary Data 8) and used to infer the maximum-likelihood phylogeny. Genome identifiers containing the GenomeID and protein accession were converted to a modified NCBI taxonomic string using an in-house script (Replace_tree_names.pl, https://github.com/ndombrowski/Phylogeny_tutorial/tree/main/Input_files/5_required_scripts). Trees were viewed in FigTree v1.4.4, and inspected for topological congruence and phylogenetic artifacts to iteratively improve the sequence selection, i.e. to exclude distant paralogs and sequences subject to long branch attraction (LBA) [133].

**Tracing the phylogenetic relationships of Eukaryotic F-Type ATP synthases**
To better resolve the evolutionary origins of eukaryotic F-type ATP synthases we constructed phylogenies with a subset of sequences which included eukaryotic sister lineages (alphaproteobacteria and cyanobacteria for the mitochondrial and plastid-type F-ATP synthase, respectively) as well as an outgroup lineage (hereafter: plastid and mitochondrial subsets). For the plastid origin dataset, we selected all eukaryotic ATP synthase sequences from the *nc*F1 and *c*F1 subunit gene trees (see above) that clustered with the Cyanobacteria and added cyanobacterial and melainabacterial homologs. Similarly, the mitochondria origin subset was generated by collecting all eukaryotic ATP synthases sequences from the *nc*F1 and *c*F1 subunit gene trees that clustered with alphaproteobacterial homologs and adding additional alphaproteobacterial sequences and gammaproteobacterial

homologs. Note that for both the plastid and mitochondrial sets, we used an expanded selection of cyanobacteria and alphaproteobacteria, respectively. Sequence selections were filtered to retain high quality sequences without ambiguous amino acids (i.e., X and x, etc.) and within the range of 450–550 bp. Closely related paralogous sequences were removed using CD-HIT v4.7 (settings: -c 0.99)[124,125] and alignments were generated using MAFFT L-INS-i v7.453[109] and trimmed using BMGE (settings: -m BLOSUM30 -h 0.55)[110]. We inferred phylogenies using the best-fit model determined in the Model Finder Plus tool in IQ-TREE2 v2.1.2 (settings: -m TESTONLY -mset LG -madd LG+C10,LG+C20,LG+C30,LG+C40,LG+C50,LG+C60,LG+C10+R+F,LG+C20+R+F,LG+C30+R+F,LG+C40+R+F,LG+C50+R+F,LG+C60+R+F --score-diff all)[111,122,123,132] and the maximum-likelihood trees were constructed in IQ-TREE v1.6.10 using the best-fit model based on the BIC[111] (Supplementary Data 8).

Additionally, we used Bayesian analysis to further verify the placement of eukaryotic F1-type ATP synthase sequences amongst the proposed sister lineages. Due to computational limitation, we downsampled the taxa subsets containing eukaryotes, alphaproteobacteria, and gammaproteobacteria to a maximum of 250 taxa (ncF1: 211, cF1:185 sequences). Sequences were cleaned, filtered, de-replicated, aligned, and trimmed using the same conditions described above. Bayesian phylogenies were constructed using PhyloBayes-MPI (version 1.5) using the CAT-GTR model with four discrete gamma categories for rates across sites; for each alignment, four independent Markov Chain Monte Carlo (MCMC) chains were run. Each chain was run over 100,000 iterations (or until convergence). Convergence was evaluated using the bpcomp and tracecomp tools within PhyloBayes-MPI, with 1000 generations discarded as burn-in and sub-sampling every 10 trees. The final consensus trees were generated through bpcomp using the same settings.

### Ancestral sequence reconstruction
Sequence alignments and the accompanying maximum-likelihood trees for the ATP synthase subunits, the orthologous pairs, and the set of four combined subunits were used to reconstruct the ancestral protein sequences. For ancestral sequence reconstruction we used a tool implemented in IQ-TREE2 v2.1.2 (settings: -m [model] -asr -te [maximum likelihood tree] -keep_empty_seq)[123]. Ancestral sequences were determined based on the proposed amino acid states at specified node positions in the rooted combined ATP synthase protein tree (Supplementary Fig. 10, Supplementary Data 7).

### Conserved nucleotide-binding motifs
Untrimmed and trimmed alignments of F- and A/V-ATP synthase subunits from the 800 reference genomes (see above) were manually inspected in Jalview v2.10.5[134] for the presence of the WalkerA (P-loop) motif[43]. The signature nucleotide-binding motif is characterized by the amino acid sequence: *GXXXXGK(T/S)* where X denotes any amino acid. The WalkerA motif sequence segment was extracted from the full alignment and used to generate conserved motif logos in WebLogo3 v3.7.4[135] (http://weblogo.threeplusone.com/).

### Dating the tree of life and ATP synthase phylogenies
The absolute time calibrations used in the dating analysis are detailed in the Supplementary Discussion. As the fossil evidence with which to constrain early microbial evolution is limited, we also used crossbracing[24] to propagate the available calibrations across the tree, implemented in McmcDate (see below). In particular, we braced the LECA node that appears in the nuclear and mitochondrial clades (setting their ages to be the same), along with all calibrated nodes within eukaryotes that were present in two or more of the eukaryotic clades (that is, we braced all nodes within eukaryotes where a geological calibration was applied). Finally, we implemented a relative

constraint[49] that the crown plastids must be younger than archaeal- and mitochondrial LECA (Supplementary Discussion).

We used McmcDate (https://github.com/dschrempf/mcmc-date) for molecular dating. McmcDate approximates the phylogenetic likelihood using a multivariate normal distribution obtained from an estimate of the posterior distribution of trees with branch lengths measured in average number of substitutions per site. We estimated the posterior distribution of trees in a previous step. For this previous step we used PhyloBayes (LG+G4 model) and a fixed phylogeny, as described above. We sampled 10,000 values of the posterior distribution of trees and observed good convergence with estimated sample size (ESS) values of around 8000.

Using McmcDate, we sampled 12,000 time trees. We used a birth-death tree prior on the time tree, and an uncorrelated log normal relaxed molecular clock model. We calibrated node ages using uniform distributions with bisected normal distributions at the boundaries. Similarly, we constrained the node order using the tails of normal distributions. We set the steepness of the boundaries individually depending on the quality and certainty of the auxiliary data. In a similar way, we used normal distributions to brace nodes. ESS values indicated good convergence and ranged from 3000 to 6000.

Application of fossil calibrations to the inferred maximum-likelihood ribosomal species tree (see above, Supplementary Fig. 20) was limited due to poor resolution of within-Eukaryote relationships. To apply an extended set of fossil calibrations we fixed the within-eukaryote topology to reflect established relationships among supergroups[136] and to allow the within-eukaryote fossil calibrations to be applied to the tree (see calibrations justification, Supplementary Discussion, Supplementary Data 18–20). In addition to these eukaryotic constraints, one topology (hereafter, Edited1) placed the nuclear eukaryotic homologs as the sister lineage to all asgardarchaeal and the mitochondrial homologs as the sister lineage to a single *Neorickettsia* (Supplementary Fig. 12). We used a more conservative approach to investigate the timing of LECA via the nuclear and mitochondrial eukaryotic nodes by adding additional constraints (hereafter, Edited2) to position the nuclear homologs as the sister lineage to the Hodearchaea (formerly Heimdallarchaeota LC3), their predicted closest asgardarchaeal relatives[4,137], and the mitochondrial homologs sister to all alphaproteobacteria, consistent with previous work[107,108] (Supplementary Fig. 13). The focal analysis described here is derived from dating the Edited2 topology (Fig. 5C, Supplementary Figs. 13 and 16).

An Approximately-Unbiased (AU) test was applied to assess the statistical support for the different topologies inferred from the ribosomal species trees utilized for cross-calibrated dating. The AU test was implemented in IQ-TREE2 v.2.1.2[123,127] (settings: iqtree2 -s 12Ribosomal_eLife_ArcBacEuk_gappyout_v1b.faa -m LG+C60+R+F -z Ribo_C60_trees.alltrees.treefile -n 0 -zb 10000 -au). Results are shown in Supplementary Data 21.

In formulating the calibrations (Supplementary Discussion), we followed the best practice principles set out in Parham et al. (2012)[138]. However, these were designed with animal and plant fossil-based calibrations and not all of the principles are applicable to calibrations of microbial clades which often lack phenotypic synapomorphies, let alone diagnostic characters that are preserved in fossil remains. Furthermore, the calibrations for many clades rely on geochemical evidence of microbial metabolisms, manifest as isotope fractionation or oxidation states of redox sensitive mineral species. Consequently, we have adapted the best practice principles to suit the nature of the calibrations. Novel calibrations are justified in full; we indicate the source of calibrations that are justified elsewhere, providing notes where they have been adapted for different clades or where the dating has changed with the revision of the geologic timescale (Supplementary Discussion).

## Gene tree-species tree reconciliation using Amalgamated likelihood estimation (ALE)

Ultrafast bootstraps (UFBoot) were inferred for each of the ATP synthase gene trees (see above) in IQ-TREE2 v2.1.2[111,122,123], and the inferred maximum-likelihood concatenated species trees (see above). ALEobserve was used to convert bootstrap distributions into ALE objects, which were reconciled using ALEml_undated against each of the four species trees: those with eukaryotes, using the LG+C20+R+F and LG+C60+R+F model, and those lacking eukaryotic sequences, with the same two models (Supplementary Data 6). These four species tree topologies were also rooted in two different ways: a root between Archaea and Bacteria, and a root between Gracilicutes and all other taxa. Two approaches were taken using ALE for gene tree-species tree reconciliation. First, we used the default ALE parameters, i.e. inferring the probability that each subunit originated at the LUCA, LBCA and LACA nodes on the prior assumption that origination at any internal node of the species tree was equally likely. We also tested an alternative approach[21] in which the origination probability at the root (O_R) is different to the origination probability for all other internal nodes of the tree, with O_R estimated by maximum-likelihood. Reconciliation analyses were performed using ALE v1.0 (https://github.com/ssolo/ALE).

To compare support for the traditional Archaea-Bacteria root for the tree of life, and an alternative root within the Bacteria, we used gene tree-species reconciliation. We performed gene tree-species tree reconciliation using the species tree as described above as well as individual gene family subunit trees of ATP synthase: ncF1 (F1 alpha), cF1 (F1 beta), cA1V1 (A1/V1 A), and ncA1V1 (A1/V1 B), as well as three combined gene families, ncF1+ncA1V1, cF1+cA1V1, and all four families combined (Supplementary Data 6). Two taxon samplings were used as described above, one with 350 Archaea and 350 Bacteria only, and another with 350 Archaea, 350 Bacteria, and 100 eukaryotes. The summed gene family likelihoods of each ATP synthase subunit were compared using an AU test[127] as implemented in CONSEL[139] under a range of conditions: species trees inferred under the LG+C20+R+F and LG+C60+R+F models; samples including and excluding eukaryotes, and two different root positions, one with the traditional root between Archaea and Bacteria, and the second with a within-Bacteria root on the branch leading to Gracilicutes.

### Reporting summary

Further information on research design is available in the Nature Portfolio Reporting Summary linked to this article.

## Data availability

All genomic data of Archaea and Bacteria analyzed are available at NCBI (Supplementary Data 1), while all eukaryotic genomic/transcriptomic material is deposited in our data repository at Zenodo (https://doi.org/10.5281/zenodo.10012837). Data generated in this study including single gene tree analyses and concatenated phylogenies (i.e., sequence files, alignments, and treefiles) have also been deposited in our data repository at Zenodo (https://doi.org/10.5281/zenodo.10012837) under the following license CC BY 4.0. Public databases are available as follows: ATP synthase Interpro domains were downloaded from Uniprot Knowledge Base (2019) (https://www.uniprot.org/), KO profiles downloaded from the KEGG Automatic Annotation Server in 2019 (https://www.genome.jp/tools/kofamkoala/), and the NCBI COG Database downloaded May 2020 (https://ftp.ncbi.nih.gov/pub/COG/COG2020/data/).

## Code availability

Workflows for annotations and phylogenies and custom R scripts to analyze and parse annotation data for figure generation have been deposited in our data repository at Zenodo (https://doi.org/10.5281/zenodo.10012837). We used the following published codes: Replace_tree_names.pl (https://github.com/ndombrowski/Phylogeny_tutorial/tree/main/Input_files/5_required_scripts), Mcmcdate (https://github.com/dschrempf/mcmc-date), catfasta2phyml.pl (https://github.com/nylander/catfasta2phyml).

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

## Acknowledgements

This work was supported by the Simons Foundation (735929LPI, to A.S., https://doi.org/10.46714/735929LPI) and the Gordon and Betty Moore Foundation (GBMF9741 to T.A.W., A.S., G.J.S., D.P. and P.C.J.D) and the Gordon and Betty Moore Foundation's Symbiosis in Aquatic Systems Initiative (GBMF9346, to A.S.). Furthermore, this project has received funding from the European Research Council (ERC) under the European Union's Horizon 2020 research and innovation programme (grant agreement No. 947317, ASymbEL to A.S. and grant agreement No. 714774, GENECLOCKS to G.J.S.). Further, this work was supported by a Royal Society University Research Fellowship to T.A.W and the John Templeton Foundation (62220, to P.C.J.D., D.P. and T.A.W). Please note that the opinions expressed in this publication are those of the author(s) and do not necessarily reflect the views of the John Templeton Foundation. We are also thankful for financial support from the Swedish Research Council (VR starting grant 2016-03559 to A.S.), the NWO-I foundation of the Netherlands Organisation for Scientific Research (WISE fellowship to A.S.). Finally, we are thankful for financial support from the Leverhulme Trust (RF-2022-167, to P.C.J.D.), the Biotechnology and Biological Sciences Research Council (BB/T012773/1, to P.C.J.D.) and the University of Bristol for a University Research Fellowship (to D.P.). We thank Gertraud Burger, Julius Lukes, Takeshi Nara, and other members of the Diplonema papillatum sequencing consortium for sharing data. We also want to thank Courtney Stairs, Andrew Roger, and Georg Hochberg for helpful discussions and/or feedback regarding eukaryotic metabolism and ancestral sequence reconstructions, respectively.

## Author contributions

A.S. and T.A.M. conceptualized the study. T.A.M., E.R.R.M., T.A.W., D.S., L.L.S., N.D., G.J.S., A.A.D., and A.S. performed analyses and interpreted data. All authors contributed methods. D.P. and P.C.J.D. contributed data. A.S., T.A.W., G.J.S and P.C.J.D. acquired funding. T.A.M. wrote the first draft with the help of A.S. and T.A.W. T.A.M., A.S., T.A.W. and E.R.R.M. together wrote the final draft. G.J.S., D.S., N.D., P.C.J.D. and D.P. contributed to the writing of the final manuscript and all authors read and approved the final version.

## Competing interests

The authors declare no competing interests.
