## [Peer Review File · Nature Communications]

ATP synthase evolution on a cross-braced dated tree of lifeReviewers' Comments:

Reviewer #1:

Remarks to the Author:

The paper by Spang, Williams and colleagues reports an extremely extensive and thorough phylogenomic and phylogenetic study of the evolution of F-type and A/V-type ATPases, in particular, their catalytic and non-catalytic (inactivated) head subunits. The work also includes absolute dating of the key events in the evolution of the ATPase subunits and life itself including LUCA, LACA, LBCA and LECA. The work is done with state of the art methods throughout.

From the viewpoint of this reviewer, this remarkable piece of work sharply splits into two major parts, the first of which is straightforward and largely non-controversial whereas the second one is daring, almost to the point of being shocking. The straightforward part includes the analysis of the distribution of the two types of ATPases across the diversity of bacteria and archaea followed by phylogeny construction and finally by the reconstruction of the key events in the evolution of ATPases subunits and linking them to the major stages in the evolution of life (origin of the LUCA, eukaryogenesis etc). The results obtained in this part of the work are quite believable and are discussed with the proper caution that is due when it comes very early events in the evolution of life. Thus, the authors lean towards a 'complex LUCA' that already encompassed both the A/V-type and the F-type ATPases. However, they duly discuss the alternative whereby the F-type ATPases that are nearly completely missing in archaea emerged from a later duplication. Furthermore, the conclusion that the split between the catalytic and the non-catalytic subunits is an ancient event antedating the LUCA is well supported and fully believable.

While most of the results and conclusions in this first part make perfect sense and appear robust, there are also some rather puzzling disturbing signs. In particular, the authors obtain the 'better' phylogeny, where among other features, eukaryotes branch from within Asgard, with non-catalytic as opposed to the catalytic subunit. The authors note non-committally that this could be due to differences in the evolutionary constraints affecting the two subunits, but this explains nothing. If anything, catalytic subunits are expected to be more strongly constrained and hence better phylogenetic markers, at least, for ancient events. The anomaly remains puzzling. Further, the phylogenetic analysis in the paper provides substantial support for the tree root inside bacteria, and although the authors finally lean towards the canonical rooting between bacteria and archaea, the support for the alternative rooting is somewhat disturbing.

The second part of the paper concerns with dating, which is obtained by "bracing" the ATPase tree with that for ribosomal proteins, and this is where the shock comes. The authors estimate an extremely ancient date for the LUCA, 4.52-4.32 Ga, whereas the divergence of the ATPase subunit antedates the LUCA (as many other ancient duplications apparently do) and is estimated to have occurred at 4.52-4.46 Ga. It has to be noted that the divergence of these subunits is a very late event in the evolution of the P-loop fold, which was undoubtedly preceded by the radiation of the major families of the P-loop enzymes (for an analysis of pre- and post-LUCA events in the evolution of the P-loop enzymes, see for example Ref. 70 in this paper). Thus, according to the dating from this paper, most of the major events in the evolution of the protein world occurred before 4.5 Ga. Now, the current estimate for the age of Earth accretion is 4.54 ± 0.05 Ga (for a supposedly authoritative discussion, see: Dalrymple, Brent G. (2004). *Ancient Earth, Ancient Skies: The Age of the Earth and Its Cosmic Surroundings*. Stanford University Press). What time does this leave for the early evolution of life? Clearly, something has to give - either the dates in this paper or those for the Earth accretion.

I am not closely familiar with the bracing approach and cannot delve into its technical details but at least, the aforementioned anomalies in phylogenies make me worry. And I certainly cannot understand why the authors do not appear to be at all concerned over the (im)plausibility of the extremely ancient dates inferred in this work.

In the opinion of this reviewer, this paper attempts to accomplish too much in one coup. Not surprisingly, the Methods section is huge and will be extremely hard for any reader to evaluate. Therefore, a reasonable solution appears to be splitting the story into two as outlined above. The phylogenomics and evolutionary reconstructions for the ATPases seem to be essentially ready for publication. The dating analysis needs to include a more thorough assessment of robustness, possibly, an attempt to validate with a different approach, and certainly, a critical discussion, taking into account the geophysical estimates of the age of Earth.

Reviewer #2:

Remarks to the Author:

In this report, the authors present a new dated tree across all life with a special focus on the timing of divergences in the history of ATP synthases. They provide interesting evidence of an early divergence between the primary lineages of ATP synthases, and evidence of the importance of horizontal gene transfer in the evolution of Eukaryotes. The study is well-executed and sheds light on early life evolution. It will be a valuable contribution, so I only have a few comments that I strongly suggest the authors address before the manuscript being considered further.

The authors mention methods that are crucial to their findings, yet these are described scantily in the main text. This would be fine if the methods were in the mainstream of phylogenetics, but that is not the case. Consider adding a description of cross-bracing in the abstract (since it is in the title), and a detailed description in the very introduction. The method is so crucial to the results, that it is unfair to leave the reader in the dark. The same is the case with Amalgamated Likelihood Estimation. Amusingly, the authors refer to the method as a 'probabilistic approach', which is exactly the only information that also appears in its name (likelihood).

The authors should consider reducing the lengths of paragraphs and sentences. Large amounts of text can be cut down, including whole sentences (e.g., lines 253-254), and text within sentences (e.g., lines 269-272). The article covers difficult topics, but the convoluted writing doubles the difficulty.

Related to this is the excessive use of acronyms. Even well-known acronyms make the text difficult to parse when used excessively (e.g., LUCA, LBACA, LACA, LECA). Other acronyms are used rarely enough that they only serve to obscure the content (e.g., HGT, AU, ALE, ASR, TOL, MAG, CPR, DPANN, AU, PP, ML, HPD, COG). At least some of these could be removed or spelled out.

Response letter to REVIEWER COMMENTS

Reviewer #1 (Remarks to the Author):

The paper by Spang, Williams and colleagues reports an extremely extensive and thorough phylogenomic and phylogenetic study of the evolution of F-type and A/V-type ATPases, in particular, their catalytic and non-catalytic (inactivated) head subunits. The work also includes absolute dating of the key events in the evolution of the ATPase subunits and life itself including LUCA, LACA, LBCA and LECA. The work is done with state of the art methods throughout.

Author response:

Thank you very much for this positive assessment.

From the viewpoint of this reviewer, this remarkable piece of work sharply splits into two major parts, the first of which is straightforward and largely non-controversial whereas the second one is daring, almost to the point of being shocking. The straightforward part includes the analysis of the distribution of the two types of ATPases across the diversity of bacteria and archaea followed by phylogeny construction and finally by the reconstruction of the key events in the evolution of ATPases subunits and linking them to the major stages in the evolution of life (origin of the LUCA, eukaryogenesis etc). The results obtained in this part of the work are quite believable and are discussed with the proper caution that is due when it comes very early events in the evolution of life. Thus, the authors lean towards a 'complex LUCA' that already encompassed both the A/V-type and the F-type ATPases. However, they duly discuss the alternative whereby the F-type ATPases that are nearly completely missing in archaea emerged from a later duplication. Furthermore, the conclusion that the split between the catalytic and the non-catalytic subunits is an ancient event antedating the LUCA is well supported and fully believable.

Author response:

We are glad to hear that the reviewer thinks that our work is remarkable and that the results obtained regarding the ATP synthase subunit evolution are valuable and adequately discussed.

While most of the results and conclusions in this first part make perfect sense and appear robust, there are also some rather puzzling disturbing signs. In particular, the authors obtain the 'better' phylogeny, where among other features, eukaryotes branch from within Asgard, with non-catalytic as opposed to the catalytic subunit. The authors note non-committally that this could be due to differences in the evolutionary constraints affecting the two subunits, but this explains nothing. If anything, catalytic subunits are expected to be more strongly constrained and hence better phylogenetic markers, at least, for ancient events. The anomaly remains puzzling. F

Author response:

We thank the reviewer for bringing up this point. In fact, while this may not have been entirely clear from the paragraph, the difference observed between catalytic and non-catalytic subunits especially relates to the placement of the eukaryotic branch relative to the Asgard archaeal ancestor in the AV-type ATP synthase phylogenies, while the placement of mitochondrial and plastid homologs was overall consistent between both subunits. We have clarified this in our revision. Furthermore, we agree that the explanation we provided is speculative at this stage (though, it is possible that too many selective constraints can result in phylogenetic artifacts such as long branch attraction because only a few mutations are accepted at a given site so that the true divergence is underestimated). However, considering that this pattern was observed in the AV but less in the F-type ATP synthases, it might be possible that the catalytic subunit of eukaryotes is difficult to place due to its functional change from an ATP synthase

to an enzyme coupling proton transport to ATP hydrolysis. We now mention both of these possibilities in the new version of the manuscript but make clear that this remains to be tested:

“This might be due to selective constraints or functional divergence considering that the eukaryotic V-type ATP synthase has evolved to couple proton transport to ATP hydrolysis rather than functioning as ATP synthase.”

Further, the phylogenetic analysis in the paper provides substantial support for the tree root inside bacteria, and although the authors finally lean towards the canonical rooting between bacteria and archaea, the support for the alternative rooting is somewhat disturbing.

Author response:

Thanks for this remark. It is evident that we were not as clear as we could have been on this point in the original manuscript, because the analysis being referred to here actually supports the “canonical” rooting of the universal tree between Archaea and Bacteria. On re-reading this passage, we realized that the phrasing was rather vague, and so we have clarified the meaning in our revision. To be clear, we obtained significantly lower likelihoods (C60, AU = 0.00009; C20, AU = 0.0002) (Supplementary Table 6) for ATP synthase subunits reconciled using a “within-bacteria” root, indicating that the root between Archaea and Bacteria is more likely. We now write:

“However, we obtained significantly lower gene family likelihoods (approximately unbiased (AU) test): C60 model, $pAU = 0.00009$; C20 model, $pAU = 0.0002$) (Supplementary Table 6) for ATP synthase subunits reconciled with a species tree rooted within Bacteria rather than between Archaea and Bacteria”

The second part of the paper concerns with dating, which is obtained by “bracing” the ATPase tree with that for ribosomal proteins, and this is where the shock comes. The authors estimate an extremely ancient date for the LUCA, 4.52-4.32 Ga, whereas the divergence of the ATPase subunit antedates the LUCA (as many other ancient duplications apparently do) and is estimated to have occurred at 4.52-4.46 Ga. It has to be noted that the divergence of these subunits is a very late event in the evolution of the P-loop fold, which was undoubtedly preceded by the radiation of the major families of the P-loop enzymes (for an analysis of pre- and post-LUCA events in the evolution of the P-loop enzymes, see for example Ref. 70 in this paper). Thus, according to the dating from this paper, most of the major events in the evolution of the protein world occurred before 4.5 Ga. Now, the current estimate for the age of Earth accretion is 4.54 +/- 0.05 Ga (for a supposedly authoritative discussion, see: Dalrymple, Brent G. (2004). Ancient Earth, Ancient Skies: The Age of the Earth and Its Cosmic Surroundings. Stanford University Press). What time does this leave for the early evolution of life? Clearly, something has to give - either the dates in this paper or those for the Earth accretion. I am not closely familiar with the bracing approach and cannot delve into its technical details but at least, the aforementioned anomalies in phylogenies make me worry. And I certainly cannot understand why the authors do not appear to be at all concerned over the (im)plausibility of the extremely ancient dates inferred in this work.

Author response:

Thanks for these remarks. We agree that our dating results are interesting and raise some intriguing questions about the pace of early evolution. That said, our results do not actually conflict with the geological evidence for the age of the Earth. In our molecular clock analyses, we implemented the Moon-forming impact (which is currently best dated to 4.52 +/- 0.05 Ga, so slightly younger than the 2004 study) as a maximum age constraint for the root of the tree. As a minimum age, we used the microfossils and geochemical evidence from 3.347 Ga rocks of the Strelley Pool Formation, which attest to the establishment of a diverse community of microbes and, potentially, metabolisms by that date. Thus, our calibrations were derived from an up-to-date interpretation of the fossil and geochemical records (see Supplementary Material for full justifications of all calibrations used). Our age estimate for LUCA is

certainly old, but with a range 4.52-4.32 Ga, it is actually younger than other recent state-of-the art analyses (e.g. Betts et al. 2018, Moody et al. 2022) and allows for hundreds of millions of years of biological evolution after the Moon-forming impact of Earth and the origin of LUCA. As discussed in the manuscript, this suggests that our cross-bracing approach actually helps to improve age estimates despite the paucity of Precambrian maximum age calibrations. We note that such early ages, while perhaps surprising *a priori*, are also consistent with the oldest fossil evidence for life: there are indications for life in the oldest rocks that could preserve evidence of life, including the (potentially) 4.280 Ga Nuvvuagittuq Supracrustal Belt in Québec (<https://www.science.org/doi/10.1126/sciadv.abm2296>), which is consistent with our results – remind that fossil evidence always postdate the origin of the lineage of interest (in this case Life itself). Our results are therefore well within the limits of plausibility, however they certainly imply a fast rate of early evolution, and in our revision we now discuss this point in more detail:

“This indicates that bracing helps to ameliorate, though not completely resolve, the problem of an under-calibrated clock inferring rates that are too low to account for the amount of genetic change that has occurred since the root of the universal tree. Some recent studies have reported moderately younger age estimates for LUCA: 4.05-3.42Ga (Fournier et al. 2021), or a range of values 4.48-3.93Ga depending on conditions (Mateos et al. 2023). An important driver of these differences is the choice of root maximum, which was younger in both studies (3.8-4.1 Ga (Mateos et al. 2023) and 3.9 Ga (Fournier et al. 2021)). In turn, also in those studies (Mateos et al. 2023; Fournier et al. 2021), the credibility interval for the age of LUCA clashes against the maximum used to calibrate the root node. This is consistent with previous work suggesting that the age of LUCA is often quite sensitive to the root calibration used (Betts et al. 2018; Parsons 2020, Moody 2022, Mateos 2023). We used the age of the Earth as our root maximum (the moon-forming impact at 4.52Ga) because we are unaware of any compelling evidence for a younger maximum on the age of extant life (Supplementary Material). Thus, while the precise age of life, and of LUCA remains uncertain, the inferred ages of LUCA and the early ATP synthase duplicates seem to imply a very high rate of evolutionary innovation during the earliest period of evolutionary history. Additional calibrations for deep nodes in the universal tree, along with estimates for other pre-LUCA paralogues, may help to dissect this key evolutionary period in higher resolution in future work (see Supplementary Discussion for further details about the resulting age estimates for major prokaryotic clades).”

Based on all the available fossil or geochemical evidence of early life our analyses and improved methods integrating all of the pertinent evidence to derive a timescale on life’s biodiversification, are the best that can currently be achieved. Our probabilistic approach represents a very significant step forward from the widespread practice of simply calibrating early evidence of life on Earth to oldest fossil evidence which is subject to recurrent claim and counterclaim.

In the opinion of this reviewer, this paper attempts to accomplish too much in one coup. Not surprisingly, the Methods section is huge and will be extremely hard for any reader to evaluate. Therefore, a reasonable solution appears to be splitting the story into two as outlined above. The phylogenomics and evolutionary reconstructions for the ATPases seem to be essentially ready for publication. The dating analysis needs to include a more thorough assessment of robustness, possibly, an attempt to validate with a different approach, and certainly, a critical discussion, taking into account the geophysical estimates of the age of Earth.

Author response:

It might be attractive to achieve a number of publications from the same study, as advocated by the referee. However, we agree with the editor in that we think the narrative is stronger in its entirety than if aspects were to be split into distinct publications. Indeed, we think that a strength of our work lies in the combination of a variety of

methods that together yield fundamental new insights into the evolution of the ATP synthase and timing of major transitions in the tree of life.

However, we agree with the reviewer that the molecular dating results can be discussed more critically and put into better perspective regarding recent work (see also above). In this light, we have added new sections addressing this aspect, discussing our results in light of other recent estimates based on different approaches, thereby validating our results.

We also agree with the reviewer that our manuscript comprises an extended/detailed method section but see this as a strength rather than a weakness: we seek to provide all necessary information to ensure reproducibility of our work. We believe that this is also in the interest of Nature Communications which strives for transparency of reporting and reproducibility of published results. Therefore, we also provide all code and accessory data in a data repository. Thus, the methods section is extensive, but this is because we aim to clearly describe a large body of technical work, including new methodological advances that are more widely applicable. We hope this level of detail will enable interested researchers to validate and further test our work. During the revisions, we have carefully edited the manuscript to make our article as clear as possible.

We hope that the reviewer agrees with this compromise.

Reviewer #2 (Remarks to the Author):

In this report, the authors present a new dated tree across all life with a special focus on the timing of divergences in the history of ATP synthases. They provide interesting evidence of an early divergence between the primary lineages of ATP synthases, and evidence of the importance of horizontal gene transfer in the evolution of Eukaryotes. The study is well-executed and sheds light on early life evolution. It will be a valuable contribution, so I only have a few comments that I strongly suggest the authors address before the manuscript being considered further.

Author response:

Thank you very much for this positive assessment.

The authors mention methods that are crucial to their findings, yet these are described scantily in the main text. This would be fine if the methods were in the mainstream of phylogenetics, but that is not the case. Consider adding a description of cross-bracing in the abstract (since it is in the title), and a detailed description in the very introduction. The method is so crucial to the results, that it is unfair to leave the reader in the dark. The same is the case with Amalgamated Likelihood Estimation. Amusingly, the authors refer to the method as a 'probabilistic approach', which is exactly the only information that also appears in its name (likelihood).

Author response:

We appreciate this view and agree that some of our analyses are not yet mainstream and have therefore added a description of both our cross-bracing approach and Amalgamated Likelihood Estimation (ALE) to the introduction; we also cite the original papers describing the logic of cross-bracing and the ALE algorithm, for readers interested in a detailed technical treatment. Further, we have added an additional sentence to the abstract regarding bracing: "We developed a phylogenetic cross-bracing approach in which we constrained corresponding nodes to the same unknown age by making use of endosymbioses and ancient gene duplications of the major ATP synthase subunits." "We developed a phylogenetic cross-bracing approach, thereby constraining equivalent speciation nodes to be contemporaneous, based on the phylogenetic imprint of endosymbioses and ancient gene duplications of the major ATP synthase subunits."

The authors should consider reducing the lengths of paragraphs and sentences. Large amounts of text can be cut down, including whole sentences (e.g., lines 253-254), and text within sentences (e.g., lines 269-272). The article covers difficult topics, but the convoluted writing doubles the difficulty.

Author response:

We have critically reread our manuscript and reduced the length of paragraphs and sentences as much as possible. Furthermore, we have removed the indicated sentences and streamlined the key messages.

Related to this is the excessive use of acronyms. Even well-known acronyms make the text difficult to parse when used excessively (e.g., LUCA, LBACA, LACA, LECA). Other acronyms are used rarely enough that they only serve to obscure the content (e.g., HGT, AU, ALE, ASR, TOL, MAG, CPR, DPANN, AU, PP, ML, HPD, COG). At least some of these could be removed or spelled out.

Author response:

In our revised version of the manuscript, we have minimized the use of acronyms as much as meaningful to improve readability of our article. Furthermore, we have written out any acronym that was used relatively infrequently (i.e. TOL, AU, ASR, MAGs, SAGs, ML, HPD).

Reviewers' Comments:

Reviewer #1:

Remarks to the Author:

I appreciate the authors' detailed and thoughtful revision and response. I cannot say that my concerns regarding both the complexity of the manuscript and the reliability of the LUCA dating are fully alleviated. However, the discussion is substantially improved (in particular, towards greater caution), and at the end of the day, publication is an editorial decision.